# Discovery of RUF6 ncRNA–interacting proteins involved in *P. falciparum* immune evasion

Gretchen M Diffendall[1,2] , Anna Barcons-Simon[1,2,5], Sebastian Baumgarten[3], Florent Dingli[4], Damarys Loew[4] , Artur Scherf[1]

**Non-coding RNAs (ncRNAs) are emerging regulators of immune evasion and transmission of *Plasmodium falciparum*. RUF6 is an ncRNA gene family that is transcribed by RNA polymerase III but actively regulates the Pol II–transcribed *var* virulence gene family. Understanding how RUF6 ncRNA connects to downstream effectors is lacking. We developed an RNA-directed proteomic discovery (ChIRP-MS) protocol to identify in vivo RUF6 ncRNA–protein interactions. The RUF6 ncRNA interactome was purified with biotinylated antisense oligonucleotides. Quantitative label-free mass spectrometry identified several unique proteins linked to gene transcription including RNA Pol II subunits, nucleosome assembly proteins, and a homologue of DEAD box helicase 5 (DDX5). Affinity purification of Pf-DDX5 identified proteins originally found by our RUF6-ChIRP protocol, validating the technique's robustness for identifying ncRNA interactomes in *P. falciparum*. Inducible displacement of nuclear Pf-DDX5 resulted in significant down-regulation of the active *var* gene. Our work identifies a RUF6 ncRNA–protein complex that interacts with RNA Pol II to sustain the *var* gene expression, including a helicase that may resolve G-quadruplex secondary structures in *var* genes to facilitate transcriptional activation and progression.**

## Introduction

Non-coding RNAs (ncRNAs) are increasingly recognized as key players in all aspects of the cellular life cycle (Cech & Steitz, 2014). ncRNAs are mainly classified into two categories, according to their length: small RNA (<200 nt) and long non-coding RNA (lncRNA) (>200 nt), although this is an arbitrary cutoff as mid-size ncRNAs ranging from 40 to 400 nt emerge as regulatory RNAs with diverse functions (Boivin et al, 2019). Mid- and long ncRNAs can play roles in diverse functions, including transcription, RNA processing, RNA degradation, and translation, while being predominantly located in the nucleus (Kopp & Mendell, 2018; Boivin et al, 2019). They are characterized by the absence of protein-coding capabilities and commonly transcribed by RNA polymerase II (Erdmann et al, 2001; Fernandes et al, 2019). Of note, the B2 RNA is transcribed by RNA Pol III, yet it can directly bind to RNA Pol II and regulate its activity (Allen et al, 2004). ncRNAs can directly interact with DNA or RNA, whereas RNA-binding proteins (RBPs) commonly contribute to regulatory functions. RBPs can stabilize ncRNAs and assist in their transport to promoter regions of genes. RBPs have also been shown to facilitate the formation of chromatin loops between distant enhancers and promoters (Postepska-Igielska et al, 2015; Cajigas et al, 2018; Tan-Wong et al, 2019; Arnold et al, 2020; Zhu et al, 2020).

The human malaria protozoan parasite *P. falciparum* remains a global health burden that claims hundreds of thousands of lives each year (W.H.O., 2021). This pathogen lacks the canonical RNAi machinery, including the key enzyme Dicer that processes transcribed shRNAs into siRNAs (Baum et al, 2009). Developmentally regulated long ncRNAs have been first described, originating from the promoter located in the intron of a virulence gene encoded by the *var* gene family (Su et al, 1995; Calderwood et al, 2003). The transcription of lncRNA originating from the *var* gene intron was observed to correlate with *var* gene activation (Jiang et al, 2013; Amit-Avraham et al, 2015; Jing et al, 2018). However, Bryant et al showed that deletion of the endogenous *var* gene intron using genome editing did not block transcriptional silencing or activation of the targeted *var* gene but did lead to higher rates of *var* gene switching (Bryant et al, 2017). Subtelomeric non-coding regions, producing long ncRNAs from repetitive DNA regions, were also reported (Broadbent et al, 2011, 2015; Raabe et al, 2012; Sierra-Miranda et al, 2012; Siegel et al, 2014). Although the biological role of most ncRNA is missing, only a few functional studies connect ncRNA mechanistically to sexual commitment and antigenic variation of malaria parasites. The activation of the Ap2G master regulator of sexual commitment is controlled by an antisense RNA of the GDV1 gene (Filarsky et al, 2018). *P. falciparum* relies on the mutually exclusive expression of virulence gene families to survive within its

[1]Universite Paris Cité, Institut Pasteur, Biology of Host-Parasite Interactions Unit, INSERM U1201, CNRS EMR9195, Paris, France   [2]Sorbonne Université Ecole doctorale Complexité du Vivant ED515, Paris, France   [3]Plasmodium RNA Biology, Institut Pasteur, Paris, France   [4]Institut Curie, PSL Research University, Centre de Recherche, CurieCoreTech Mass Spectrometry Proteomics, Paris, France   [5]Biomedical Center, Division of Physiological Chemistry, Faculty of Medicine, Ludwig-Maximilians-Universität München, Munich, Germany

Correspondence: artur.scherf@pasteur.fr

host (reviewed in Guizetti and Scherf [2013] and Wahlgren et al [2017]). In particular, the 60-member *var* multigene family codes for PfEMP1, an important variant surface adhesion molecule central to the development of the pathogenesis of the disease (Miller et al, 2002). All *var* genes are composed of a 5′ upstream promoter followed by exon I, a relatively conserved intron, and exon II at subtelomeric and central locations in each chromosome. Transcription of this gene family is tightly regulated by multiple epigenetic layers to ensure mutually exclusive expression at a specific expression site. Remarkably, G4-quadruplex motifs are enriched in *var* gene promoter and coding regions (Smargiasso et al, 2009; Gage & Merrick, 2020; Gazanion et al, 2020). The stabilized coding-strand G4 can inhibit transcription of reporter genes in transfected *P. falciparum* (Harris et al, 2018), raising the possibility that specific helicases may be involved in *var* gene activation (Smargiasso et al, 2009).

The active *var* gene is enriched in histone marks H3K4me3 and H3K9ac, and in the histone variant H2A.Z, and localizes to a distinct perinuclear expression site (Ralph et al, 2005; Lopez-Rubio et al, 2007, 2009; Petter et al, 2011). All other *var* genes remain transcriptionally silenced, and tethered in repressive clusters enriched in histone modifications H3K9me3 and H3K36me3, and heterochromatin protein 1 (Pf-HP1) and at the nuclear periphery (Flueck et al, 2009; Lopez-Rubio et al, 2009; Pérez-Toledo et al, 2009; Jiang et al, 2013). Although the histone mark signatures for active, poised, and silent *var* genes have been described (Guizetti & Scherf, 2013), it is unclear how the RNA Pol II transcriptional machinery is recruited to the *var* gene expression site.

The RUF6 ncRNA gene family (15 members) has recently been proposed to be linked to the mutually exclusive transcription of the *var* multigene family (Guizetti et al, 2016; Wu et al, 2019; Barcons-Simon et al, 2020). RUF6 ncRNA was shown to colocalize in *trans* to the *var* gene expression site by FISH, and the episomal overexpression of RUF6 resulted in the up-regulation of previously silenced *var* genes, thus disrupting monoallelic expression (Guizetti et al, 2016). Down-regulating the entire RUF6 gene family, using CRISPR interference, led to a concurrent down-regulation in all *var* genes (Barcons-Simon et al, 2020). RUF6 genes are 135 nt long and have a GC content higher than 50%, which is striking when compared to ~20% GC content genome-wide (Gardner et al, 2002; Chakrabarti et al, 2007). Interestingly, this ncRNA gene family is conserved only in *Plasmodium* species from the *Laverania* subgenus that encode a *var* gene family (Otto et al, 2018). Unlike other ncRNAs, RUF6 members contain canonical A- and B-box required for RNA polymerase III (Pol III) transcription. Although the RUF6-mediated *var* gene activation has been genetically validated, an understanding of how nuclear RUF6 ncRNA–interacting proteins mechanistically connect to their downstream effectors is missing.

In this study, we shed light on the mode of action of the RUF6 ncRNA by exploring its interaction partners. To identify specific RUF6 ncRBPs in their native cellular context, we used a recently reported method termed Chromatin Isolation by RNA Purification followed by liquid chromatography–mass spectrometry (ChIRP-MS) (Chu et al, 2015; Chu & Chang, 2018). Chu and colleagues first used the technique to newly identify novel protein partners of the lncRNA Xist, essential for X chromosome inactivation in mammals. We adapted ChIRP-MS to *P. falciparum* for the first time, which allowed for the systematic discovery of RUF6 ncRBPs. The identified

interactome indicates that RUF6 ncRNA interacts directly with RNA Pol II at the *var* expression site. Using RNA immunoprecipitation (RIP), proteomic, and an inducible knock-sideways approach, we identify the DEAD box RNA helicase, Pf-DDX5, as a regulator of transcription of virulence genes.

# Results

### Identification of RUF6 ncRNA–interacting proteins by ChIRP-MS

Given the lack of sequence homology between *var* gene loci and RUF6 ncRNA, we hypothesize that RUF6 ncRNA associates with the active *var* loci via RBPs (Fig 1A). To test this hypothesis, we conducted electrophoretic mobility shift assays using a biotinylated probe of an ncRNA RUF6 member (PF3D7_1241000). We observed a specific gel shift when incubating the probe with a nuclear extract of asynchronous parasites that was reproducible among four independent experiments. An excess of an identical unlabeled probe competed for the binding factor, resulting in lack of a shift in the labeled probe. The same result was not obtained if a non-specific competitor probe was used (Fig 1B). These data suggest the presence of a specific RUF6 ncRNA binding factor.

To identify potential RUF6 ncRBPs in vivo, we adapted the Chromatin Isolation by RNA Purification (ChIRP) method developed by Chu et al (Chu et al, 2012, 2015; Chu & Chang, 2018) to *P. falciparum* (Fig 1C). This technique cross-links cells in their native context using formaldehyde before hybridization of the target RNA with biotinylated oligonucleotides. Biotin-associated proteins are digested on-beads and identified after peptide desalting by liquid chromatography–tandem mass spectrometry (LC-MS/MS). Negative controls such as non-interacting probes and RNase A treatment of lysate before ChIRP are included.

To investigate the interactome of RUF6, we performed ChIRP-MS on a 3D7 clone, which expresses the RUF6 gene PF3D7_0412800 and the adjacent *var* gene PF3D7_0412700 (Barcons-Simon et al, 2020). We designed two sets of antisense biotinylated (3′ TEG–biotin) DNA oligo probes (odd and even) that hybridize to the active ncRNA and target the regions less prone to be located within a loop according to the predicted secondary structure (Fig 1D and Table S1). As a control, we used a pool of two different scrambled probes that had similar GC content to the targeted probes but were not predicted to bind anywhere in the genome. To eliminate non–RNA-specific protein interaction partners, one additional experimental group was defined in which cells were first treated with RNase A, then incubated with RUF6-targeted probes. Quantitative reverse transcription–PCR (RT–qPCR) on RNA isolated from each ChIRP replicate showed that the RUF6 ncRNA was specifically enriched with both targeted odd and even probes, but not with the scrambled probes or the samples treated with RNase A (Fig 1E).

Having demonstrated the specificity of the ChIRP technique with RUF6 ncRNA, we performed ChIRP and on-bead digestion followed by LC-MS/MS at 18 hpi, when both the active *var* and RUF6 ncRNA are transcribed, to identify proteins that interact with RUF6 ncRNA in vivo. We used four biological replicates, grown and harvested at separate times, and performed a label-free mass spectrometry quantification to identify enriched proteins in the targeted even

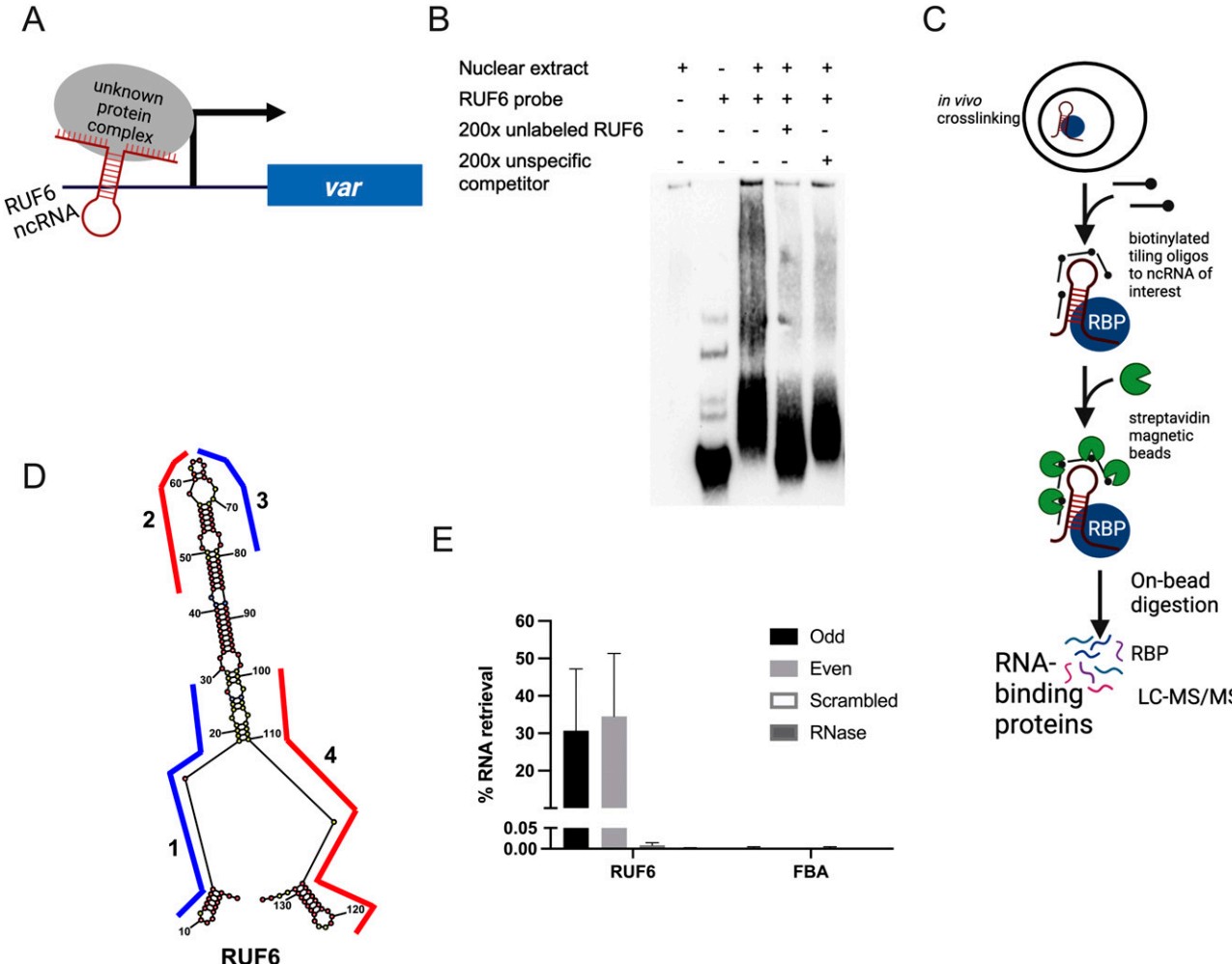

**Figure 1. Developing ChIRP-MS to identify RUF6 ncRNA–binding proteins.**
**(A)** Schematic showing the hypothetical model of RUF6 ncRNA associating with proteins at the active *var* gene promoter. **(B)** Electrophoretic mobility shift assay using a biotinylated RUF6 ncRNA probe (Pf3D7_1241000) gives a specific shift in the presence of nuclear extract that is competed for using an unlabeled probe in excess (200×). **(C)** Outline of the ChIRP-MS protocol. RNP complexes are cross-linked in vivo before addition of biotinylated antisense oligos. Target ncRNAs are pulled out with beads, and RNA-binding proteins are identified through LC-MS/MS. Created with BioRender.com. **(D)** Lowest free energy secondary structure of the RUF6 coded by PF3D7_0412800, as predicted by RNA structure bioinformatic web server. The four oligonucleotide probes used for ChIRP-MS are shown along their binding regions of the ncRNA structure. Odd probes 1 and 3 are colored in blue, and even probes 2 and 4 are colored in red. **(E)** Percentage of RNA retrieval in ChIRP-MS compared with input samples using odd, even, and scrambled sets of probes and RNase-treated samples. Transcript levels were assessed by RT–qPCR, and fructose–bisphosphate aldolase (PF3D7_1444800) levels were used as a negative control.

probe samples compared with the control samples. This led to the quantification of 571 total *P. falciparum* proteins (false discovery rate [FDR] of 1% and the number of proteotypic peptides used ≥ 3; Figs 2A and S1 and Table S2). Target probes were compared with each of the two controls separately to determine a list of candidate proteins. We selected proteins that were significantly enriched or unique in the target sample compared with one or both controls (ratio ≥ 2.0, adjusted $P ≤ 0.05$ or unique proteins). The targeted RUF6 probes for all four replicates yielded a total of 386 significantly enriched or unique proteins (Fig 2A and Tables S3 and S4). Unique proteins in the target samples included three ApiAP2 transcription factors, two subunits of RNA Pol II, chromodomain–helicase–DNA-binding protein (Pf3D7_1023900.1), DNA topoisomerase 2 (Pf3D7_1433500.1), transformer-2 (Pf3D7_1002400.1), ATP-dependent RNA

helicase DDX5 (Pf3D7_1445900.1), and multiple RBPs and proteins with unknown functions. An RBP (Pf3D7_1330800) was significantly enriched in target samples compared with both control groups (ratio = 4.92, adj. $P = 9.44 × 10^{-3}$ versus scrambled probes; and ratio = 3.73, adj $P = 1.49 × 10^{-5}$ versus RNase-treated samples). Significantly enriched proteins in target samples compared with control scrambled probe samples included the following: a DNA/RBP Alba2 (Pf3D7_1346300, ratio = 12.13, $P = 0.05$) and proteins with unknown functions (Pf3D7_0813300, ratio = 5.71, $P = 6.81 × 10^{-4}$; and Pf3D7_0821400, ratio = 3.74, $P = 0.02$). Proteins significantly enriched or uniquely found with targeted probes gave Gene Ontology terms related to mRNA ($P = 9.96 × 10^{-20}$), RNA ($P = 1.14 × 10^{-18}$), and nucleic acid ($P = 1.87 × 10^{-15}$) binding (Fig 2C and Table S5). Such terms are consistent with transcription and gene regulation.

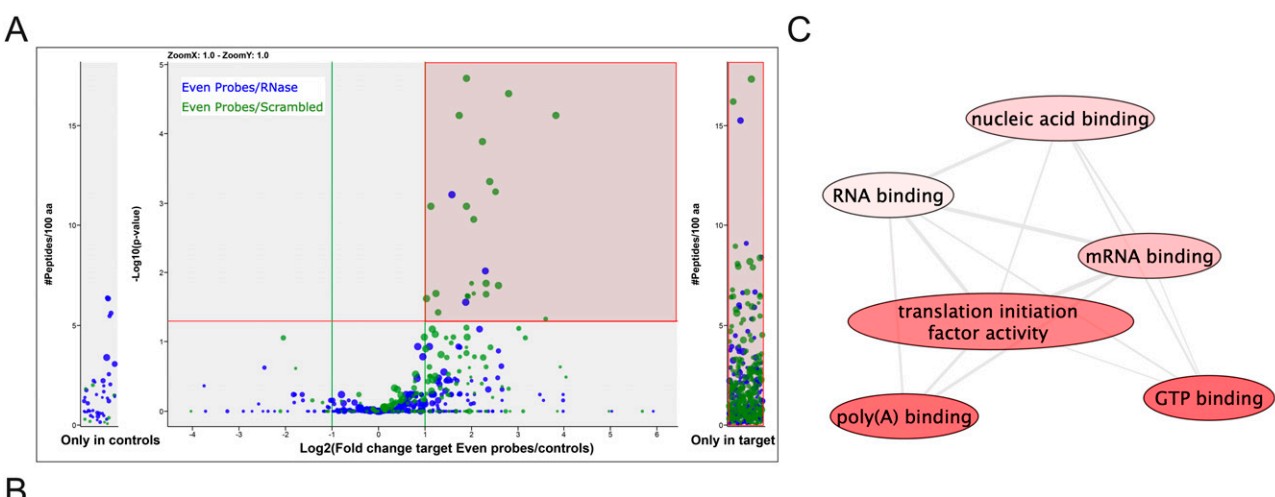

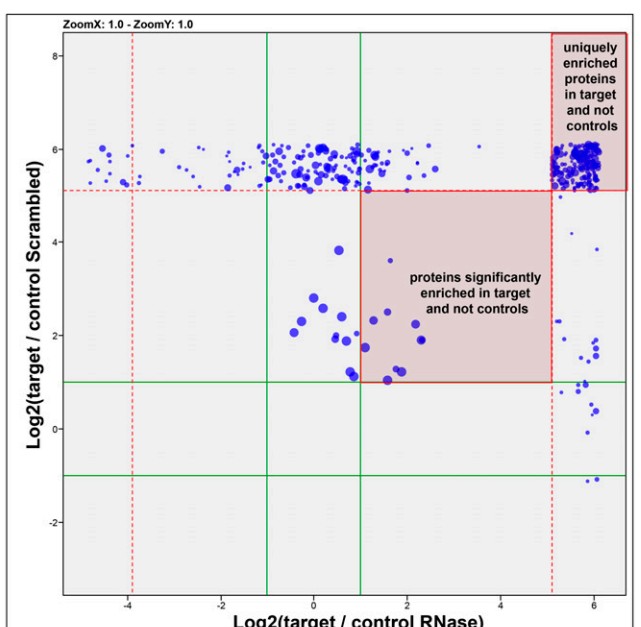

| Proteins | MW (kDa) | Description |
|---|---|---|
| PF3D7_0318200 | 278,7 | DNA-directed RNA polymerase II subunit RPB1 |
| PF3D7_0605100 | 85,6 | RNA-binding protein |
| PF3D7_1423700 | 183,6 | conserved Plasmodium protein, unknown function |
| PF3D7_1445900 | 60,0 | ATP-dependent RNA helicase DDX5 |
| PF3D7_1023900 | 381,3 | chromodomain-helicase-DNA-binding protein 1 |
| PF3D7_1104200 | 167,4 | chromatin remodeling protein |
| PF3D7_1002400 | 30,1 | transformer-2 protein |
| PF3D7_0819600 | 32,2 | conserved Plasmodium protein, unknown function |
| PF3D7_0923900 | 23,0 | RNA-binding protein |
| PF3D7_1007700 | 182,7 | AP2 domain transcription factor AP2-I |
| PF3D7_1107300 | 381,9 | polyadenylate-binding protein-interacting protein 1 |
| PF3D7_1110200 | 152,2 | pre-mRNA-processing factor 6 |
| PF3D7_1330800 | 68,0 | RNA-binding protein |
| PF3D7_1455300 | 69,5 | conserved Plasmodium protein, unknown function |
| PF3D7_0919000 | 31,8 | nucleosome assembly protein |

**Figure 2. ChIRP-MS identification of RUF6 ncRNA–interacting proteins.**
**(A)** RUF6 ChIRP-MS volcano plot of label-free quantitative proteomic analysis of 571 *P. falciparum* proteins present for all four replicates in target samples (even probes) compared with the control groups: scrambled probes and RNase-treated samples. The blue dot color represents target (even probes) versus control (RNase-treated), and the green dot color represents target (even probes) versus control (scrambled probes) quantifications. Each dot represents a protein, and its size corresponds to the sum of peptides from both conditions used to quantify the ratio of enrichment. $x$-axis = $\log_2$(fold change), $y$-axis = $-\log_{10}(P$-value), horizontal red line indicates adjusted $P$ = 0.05, and vertical green lines indicate absolute fold change = 2.0. Side panels indicate proteins uniquely identified in either sample ($y$-axis = number of peptides per 100 amino acids). All individual comparisons can be found in supplementary figures. Red highlighted boxes show 386 proteins significantly enriched or unique in target samples compared with each control. **(A, B)** Correlation plot comparing target (even probes) versus control (RNase-treated), and target (even probes) versus control (scrambled probes) of the 386 proteins from (A). Red highlighted boxes show proteins significantly enriched or unique in target samples compared with both controls. **(C)** Molecular function groupings displayed represent Gene Ontology terms related to the molecular function of genes enriched only in RUF6 target samples but not in control samples, prepared for visualization using PlasmoDB (plasmodb.org) and the REViGO tool (http://revigo.irb.hr/). Proteins had a $P$-value cutoff at 0.05. **(D)** Protein IDs of selected candidate proteins.
Source data are available for this figure.

## Analysis of ChIRP-MS–interacting proteins

For downstream analysis, we selected candidate proteins if they met one or more of the following selection criteria: (1) have predicted RNA-binding potential, (2) have a function in gene activation, and/or (3) are conserved specifically in the *var* gene–containing

*Laverania* species but not in malaria species that do not encode *var* genes (*Plasmodium* species). This selection left a list of 15 candidates: 11 unique proteins in the target sample compared with both controls and four unique proteins in the target sample compared with the control scrambled probe sample (Fig 2B and D and Tables S6 and S7). The top six candidates were selected to begin with

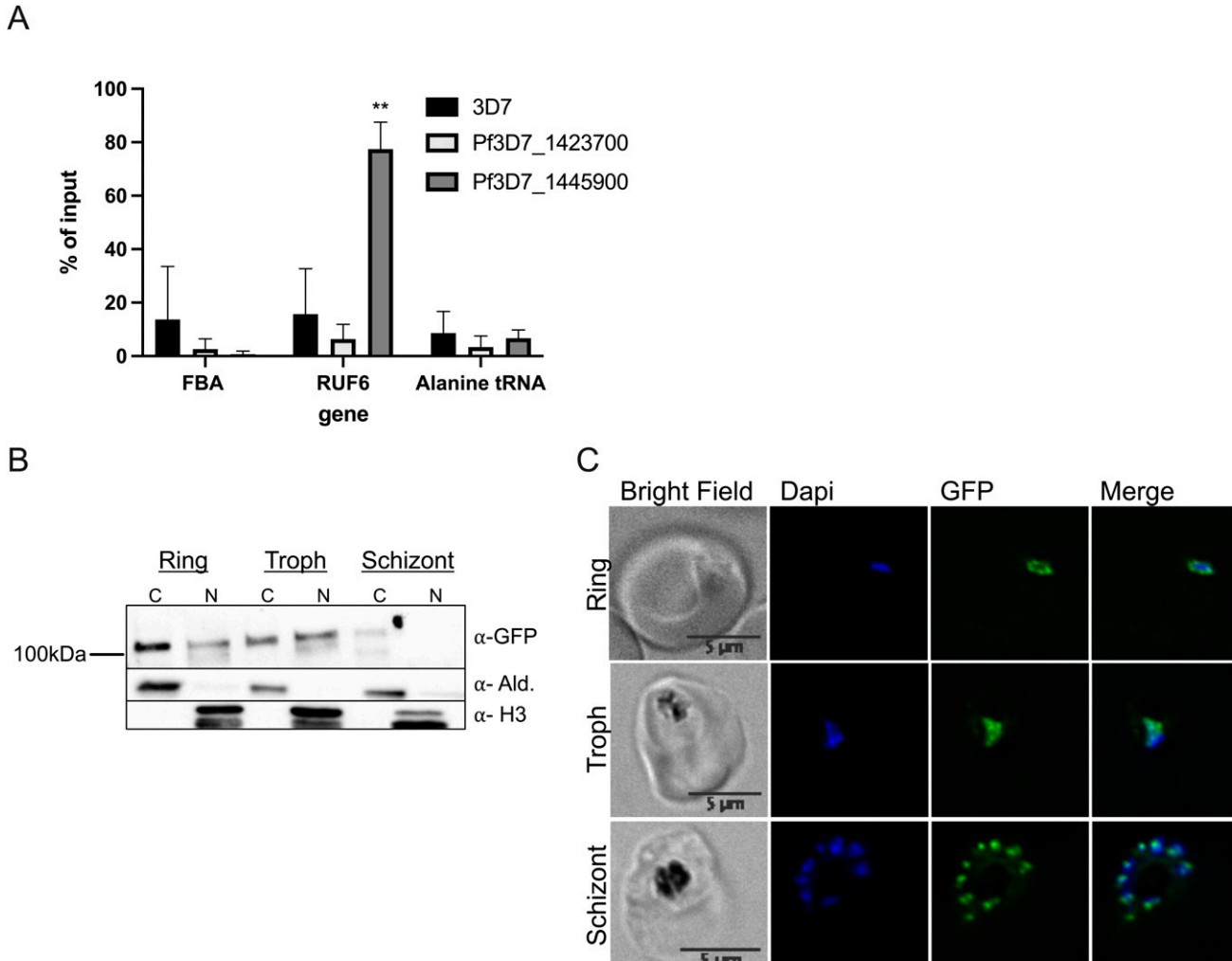

**Figure 3. Validation of ChIRP-MS candidate proteins.**
**(A)** RT–qPCR results from RNA immunoprecipitation analysis on WT 3D7 nuclear extracts: protein with unknown functions (Pf3D7_1423700) and Pf-DDX5 RNA helicase (Pf3D7_1445900). Primers used for RT–qPCR were fructose–bisphosphate aldolase (FBA), Pf3D7_1444800; RUF6 for the entire gene family; and alanine tRNA, Pf3D7_0411500. Results are displayed as % of input. Error bars are displayed from three biological replicates. **(B)** Western blot of Pf-DDX5 in cytoplasmic and nuclear extracts throughout the IDC (rings, trophs, and schizonts). **(C)** Representative immunofluorescence images show bright field, DAPI, GFP, and DAPI-GFP merge for candidate protein Pf-DDX5.

further validation because of their predicted RNA-binding properties and involvement in gene regulation (Fig 2D). We were successful in C-terminal tagging of two of the six candidates for further validation. We used the selection-linked integration (SLI) strategy (Birnbaum et al, 2017) to generate strains in which the candidate proteins Pf3D7_1445900 Pf-DDX5 and Pf3D7_1423700 with unknown functions were tagged with a GFP epitope. To confirm that the candidate proteins bind to RUF6 ncRNA, we performed RIP followed by RT–qPCR with three biological replicates, grown and harvested at separate times, for WT 3D7 parasites and the GFP-tagged proteins of interest in nuclear extracts of ring-stage parasites. A significant enrichment ($P$ = 0.0078) was observed for RUF6 in Pf-DDX5-GFP parasites, but not in Pf3D7_1423700-GFP parasites (Fig 3A). No enrichment was observed for controls mRNA fructose–bisphosphate aldolase (FBA, Pf3D7_1444800) or alanine tRNA (Pf3D7_0411500) (Fig 3C). These data suggest that the Pf-DDX5, but not the other candidate

protein (Pf3D7_1423700), binds to RUF6 ncRNA. Western blot (Fig 3B) and immunofluorescence (Fig 3C) analyses showed that the Pf-DDX5 was present in the nucleus in ring and trophozoite stages, during which the *var* gene and *RUF6* transcription peaks. Having demonstrated the association between Pf-DDX5 and RUF6 ncRNA, we set out to gain insight into the protein interactome of Pf-DDX5. We performed immunoprecipitation followed by quantitative mass spectrometry of Pf-DDX5-GFP and GFP-tagged Pf3D7_1423700, which served as a control because it was found to not bind RUF6 by RIP (Fig 3A). A total of five biological replicates, with the parasites grown and harvested at separate times, were prepared. Analysis of the quantitative mass spectrometry data revealed a significant enrichment of the control protein (ratio = 3.00, $P$ = 1.13 × 10$^{-14}$) and Pf-DDX5 (ratio = 4.88, $P$ = 3.92 × 10$^{-28}$) in their respective samples (Fig 4A and Table S8).

A comparison between the proteins that were significantly enriched in the Pf-DDX5 IP-MS and those in the RUF6 ChIRP-MS

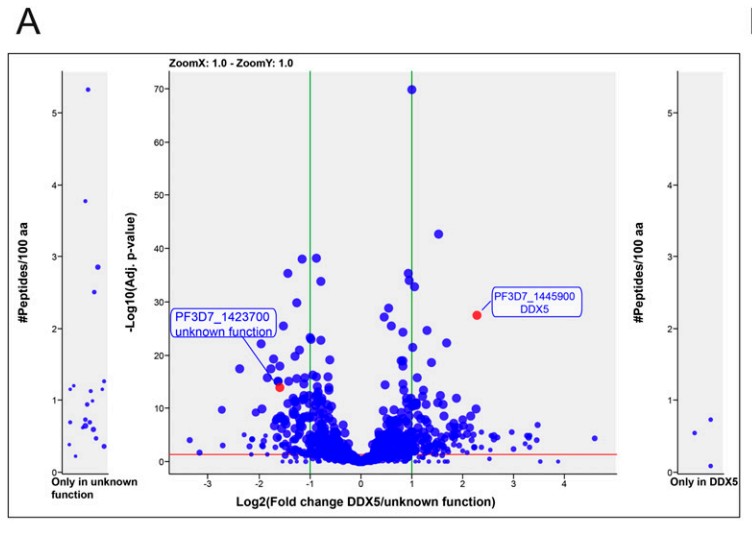

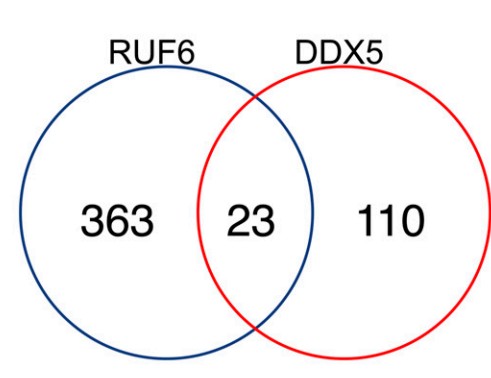

**Figure 4. Pf-DDX5 interactome.**
**(A)** Co-IP-MS volcano plot of enrichment for all five replicates for Pf-DDX5 versus control protein, a protein with unknown functions (Pf3D7_1423700); proteins are indicated and labeled. Each dot represents a protein, and its size corresponds to the sum of peptides from both conditions used to quantify the ratio of enrichment. x-axis = $\log_2$(fold change), y-axis = $-\log_{10}$(P-value), horizontal red line indicates adjusted $P$ = 0.05, and vertical green lines indicate absolute fold change = 2.0. Side panels indicate proteins uniquely identified in either sample (y-axis = number of peptides per 100 amino acids) with a minimum of three total peptides. Adjusted $P$-value of 0.05 is displayed as a horizontal red line, and fold change greater than 2 is labeled as vertical green lines. **(B)** Venn diagram showing a total number of shared proteins found in RUF6 ChIRP-MS and DDX5 IP-MS. **(C)** Biological process groupings displayed represent shared significant proteins from RUF6 ChIRP-MS and DDX5 IP-MS prepared for visualization using PlasmoDB (plasmodb.org) and the REViGO tool (http://revigo.irb.hr/). Proteins had a $P$-value cut-off at 0.05. **(D)** Significant proteins enriched in the Pf-DDX5 sample that are common to the original ChIRP-MS selected candidate proteins.
Source data are available for this figure.

revealed key similarities and differences. 110 proteins were significantly enriched only in the Pf-DDX5 IP-MS, but not in the RUF6 ChIRP-MS, suggesting that DDX5 could be a part of multiple protein complexes with various cellular functions not limited to RUF6-mediated *var* gene regulation. In fact, although the GO analysis of all significantly enriched proteins in the Pf-DDX5 sample showed the top results for translation ($P$ = 1.90) and the structural component of the ribosome ($P$ = 1.36; Tables S9 and S10), these changed greatly when only looking at shared proteins in the Pf-DDX5 sample that were also found in the targeted RUF6 ChIRP-MS. 23 proteins that were unique or significantly enriched in the RUF6 ChIRP-MS

were also significantly enriched in the DDX5 IP-MS (Fig 4B and Table S11).

These shared proteins have functions mainly related to regulation of DNA-templated transcription initiation by RNA Pol II ($P$ = $1.27 \times 10^{-4}$; Fig 4C and Table S12). The GO analysis of these shared proteins revealed that three of the topmost significantly enriched molecular functions were protein binding ($P$ = $4.49 \times 10^{-6}$), RNA binding ($P$ = 0.006), and single-stranded DNA exodeoxyribonuclease activity ($P$ = 0.009; Table S13). The biological process of GO analysis showed the chromatin assembly ($P$ = $2.16 \times 10^{-3}$) and two of the four proteins having to do with "regulation of transcription initiation

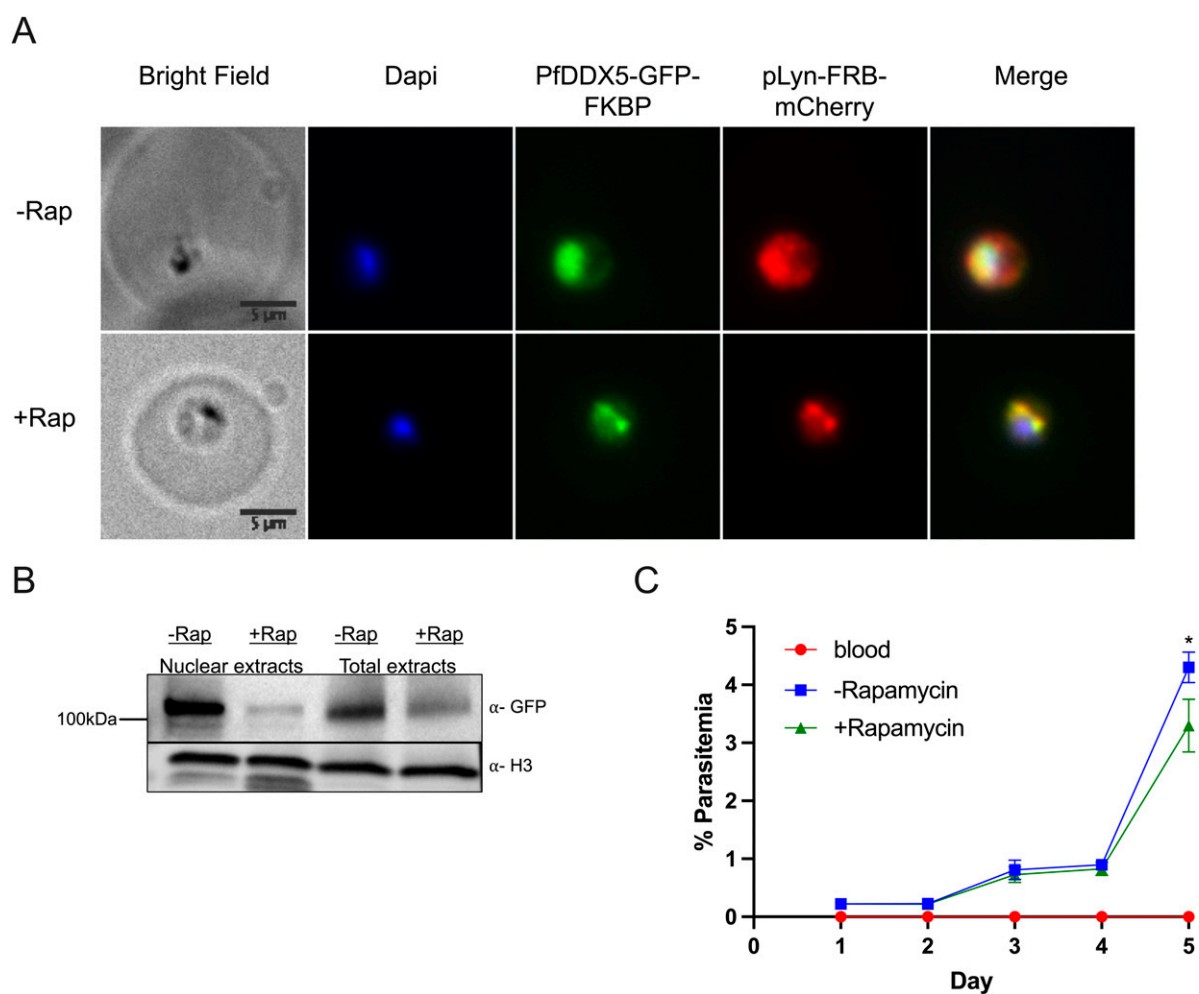

**Figure 5. Knock-sideways of Pf-DDX5.**
**(A, B)** IFA images of Pf-DDX5-GFP-2FKBP mislocalization with plasma membrane mislocalizer, pLyn-FRB-mCherry, after addition of rapamycin. (B) Western blot displaying Pf-DDX5 protein removal from the nucleus after addition of rapamycin. Histone H3 was used as a nuclear extract control. Using image analysis tools, addition of rapamycin caused a 10× decrease in nuclear levels (normalized to H3). Total extracts show Pf-DDX5 presence before and after addition of rapamycin. **(C)** Growth curve over 4 d of Pf-DDX5-GFP-2FKBP parasites cultured in the presence (250 nM final concentration, +rapamycin) or absence (–rapamycin) of rapamycin. Error bars represent SDs from three independent experiments.

from RNA polymerase II promoter" and "positive regulation of RNA polymerase II transcription preinitiation complex assembly" were found in the shared proteins between RUF6 and Pf-DDX5 samples ($P = 1.27 \times 10^{-4}$; Table S12).

Importantly, seven proteins from our original shortlisted candidate proteins from the RUF6 ChIRP-MS (Fig 2D) were found to be significantly enriched in the Pf-DDX5 IP-MS: proteins with unknown functions (Pf3D7_1455300.1, ratio = 2.34, $P = 1.99 \times 10^{-4}$; Pf3D7_0721100.1, ratio = 1.99, $P = 2.51 \times 10^{-6}$; and Pf3D7_0819600.1, ratio = 2.05, $P = 6.44 \times 10^{-4}$), polyadenylate-binding protein (Pf3D7_1107300.1, ratio = 3.14, $P = 3.95 \times 10^{-5}$), chromatin remodeling protein (Pf3D7_1104200.1, ratio = 6.39, $P = 2.59 \times 10^{-5}$), pre-mRNA–processing factor (Pf3D7_1110200.1, ratio = 2.39, $P = 3.39 \times 10^{-2}$), and nucleosome assembly protein (Pf3D7_0919000, ratio = 2.03, $P = 1.72 \times 10^{-3}$). Finding shared proteins between the two techniques further strengthens the evidence that Pf-DDX5 is a part of a larger protein complex associated with RUF6 ncRNA. Shared proteins have GO

analysis results for nucleosome positioning, chromatin organization, and protein–DNA complex subunit organization ($P = 1.23 \times 10^{-3}$, $1.84 \times 10^{-2}$, and 0.04).

### Nuclear DDX5 is involved in *var* gene regulation

We hypothesized that RUF6 ncRNA–associated Pf-DDX5 may have a direct role in the immune evasion mechanism. We used a knock-sideways approach (described in Birnbaum et al [2017]) to study the function of Pf-DDX5. Knock-sideways is based on the ligand-induced dimerization of FK506-binding protein (FKBP), which is fused to the endogenous Pf-DDX5, and FKBP–rapamycin-binding protein (FRB), which is separately expressed and fused to a mis-localizer, anchored to the parasite plasma membrane (PPM). Upon addition of rapamycin, Pf-DDX5 levels decreased substantially in the nucleus (60-h treatment) (Fig 5B), and colocalization with the FRB mislocalizer increased in the PPM (Fig 5A). The parasite growth

appeared to be unaffected after two growth cycles, but lower parasitemia was observed after 5 d of treatment ($P$ = 0.031) (Fig 5C), indicating a fitness cost for Pf-DDX5 displacement. This observation supports the phenotype observed for piggyBac transposon mutagenesis for this gene (Zhang et al, 2018). To determine the effects of Pf-DDX5 knock-sideways on transcription, we treated synchronized parasites with rapamycin for 60 h, which is within a window of time when growth should not yet be affected, and harvested ring-stage parasites when *var* gene transcription reaches its peak. Giemsa staining of the harvested parasites confirms that both control and rapamycin-treated parasites were at the ring stage (Fig S2). In addition, the age of the parasites was estimated to be the ring stage (Fig S3), shown in a heatmap of Pearson's *r* correlation coefficients. Three biological replicates were used with the parasites grown and harvested at separate times. Rapamycin-induced mislocalization of Pf-DDX5 resulted in significantly down-regulated (568) and up-regulated (301) genes (Benjamini–Hochberg-adjusted *P*-value (i.e., *q*) was found to be ≤ 0.05) (Table S14).

RUF6 transcript levels were not affected, indicating that Pf-DDX5 does not interfere with the stability of this ncRNA (Fig 6A). Importantly, the single active *var* gene in this clone was one of the topmost significantly down-regulated genes ($q$ = $2.59 \times 10^{-6}$) (Fig 6B–D).

GO enrichment analysis revealed that down-regulated genes were enriched in the biological processes of translation ($P$ = $9.47 \times 10^{-34}$) and RNA processing ($P$ = $9.69 \times 10^{-22}$; Table S15). Genes included in these categories encode splicing factors, a subunit of RNA Pol III, and proteins with unknown functions. GO enrichment analysis revealed that up-regulated genes were enriched in the biological processes: movement in host environment ($P$ = $2.41 \times 10^{-21}$) and entry into host ($P$ = $4.75 \times 10^{-21}$; Table S16). Together, these data indicated that the recruitment of Pf-DDX5 by RUF6 ncRNA to the active *var* gene promotes transcription, possibly by resolving secondary structures that are highly enriched in this gene family (Gazanion et al, 2020).

## Discussion

Despite the association between ncRNAs as regulators of malaria parasite pathogenesis and transmission, the underlying molecular mechanisms remain elusive. To overcome technical challenges and explore key open-ended questions, we developed a new tool for malaria parasites aimed to identify ncRNA–protein interactions in their native cellular context. We established a robust ChIRP-MS protocol using biotinylated tiling oligo probes for RUF6 (see Fig 1D). After validating our probes in pulldown assays, this new tool was used to identify RUF6-associated proteins. Importantly, the ChIRP-MS method allowed us to isolate and identify specific RUF6 RNA–associated proteins that were not detected using nuclear extracts in RUF6 ncRNA affinity purification assays (unpublished data). This result highlights that the "native" ChIRP-MS method detects interactions that cannot be reconstituted once the nucleus is disrupted. This method may find broad applications in the malaria field to identify specific RBPs.

The rationale of this study was to gain biological insight into the RUF6 ncRNA–mediated activation of the *var* gene family, which encodes a major virulence factor and contributes to immune evasion from the host immune response. The RUF6 gene family is strictly linked to a small subset of malaria species that encode the *var* virulence gene family (Otto et al, 2018). This evidence has motivated previous studies to explore its role in the monoallelic expression of the *var* gene family (Wei et al, 2015; Guizetti et al, 2016; Barcons-Simon et al, 2020). It remains unknown how the ncRNA RUF6 is targeted in *trans* to an active *var* gene expression site and which proteins RUF6 recruits and interacts with to promote efficient singular *var* gene transcription. Here, we established a comprehensive RUF6 ncRNA protein interactome and validated the helicase DDX5 as a regulator of *var* gene transcription. Two controls were used (scrambled probes and RNase A–treated samples) to raise the prospect of identifying protein candidates that either bind directly or are associated with RUF6 ncRNA. We pulled out binding proteins that were detected with label-free quantitative proteomics (LC-MS/MS) (Fig 2A). One challenging aspect of proteomic analysis is the amount of input material needed to overcome non-specific background hits. To overcome this obstacle, we used $4.0 \times 10^{10}$ parasites in ChIRP-MS (n = 4) (and later $1.5 \times 10^9$ parasites in Co-IP-MS, n = 5). Quantitative analysis showed more proteins that were uniquely found in our RUF6 target samples rather than proteins more statistically enriched in our target samples compared with controls ($P$ < 0.05). Therefore, our candidate proteins were uniquely identified in our target samples and absent in one or both control groups. Among those candidates were genes that encode proteins implicated in gene transcription and candidates with unknown functions. We hypothesized that some proteins bind specifically to the ncRNA and others form complexes via protein–protein interactions.

Candidates involved in transcription are homologous to two subunits of RNA Pol II, to DNA topoisomerase 2, to DDX5 RNA helicase, to nucleosome assembly proteins, and to CHD1, a chromatin remodeling protein. These data point to an interaction of the RUF6 complex with the *var* gene RNA Pol II transcriptional machinery. It is noteworthy in this context that RNA Pol II transcribes *var* genes, but RUF6 is transcribed by RNA Pol III. In fact, an ncRNA transcribed by RNA Pol III (called B2 RNA) has been shown to act in *trans* to bind directly to RNA Pol II to regulate transcription in mice (Espinoza et al, 2004). Notably, two candidate proteins have been previously reported to bind to or near the *var* gene promoters, Alba DNA/RBPs (*Archaeal* chromatin protein family) and a member of the ApiAp2 transcription factor family (Chêne et al, 2012; Goyal et al, 2012; Martins et al, 2017). In addition, the candidate protein transformer-2, not yet characterized in *Plasmodium*, is known to act in insects as an upstream regulatory element in sexual regulation (Nguyen et al, 2021). Furthermore, the polyadenylate-binding protein was among the candidate proteins and has been found in *Drosophila* to interact with RNA Pol II at the promoter during transcription (Kachaev et al, 2019). These findings corroborate our previously proposed mode of function, namely the colocalization of RUF6 ncRNA with actively transcribed *var* genes (Guizetti et al, 2016), and that the RUF6 transcription is linked to *var* gene activation (Guizetti et al, 2016; Barcons-Simon et al, 2020). We performed functional validation of one of the identified interactome proteins, which is predicted to bind to RNA. It is a homolog of the human DEAD box protein DDX5, an RNA helicase, that can also

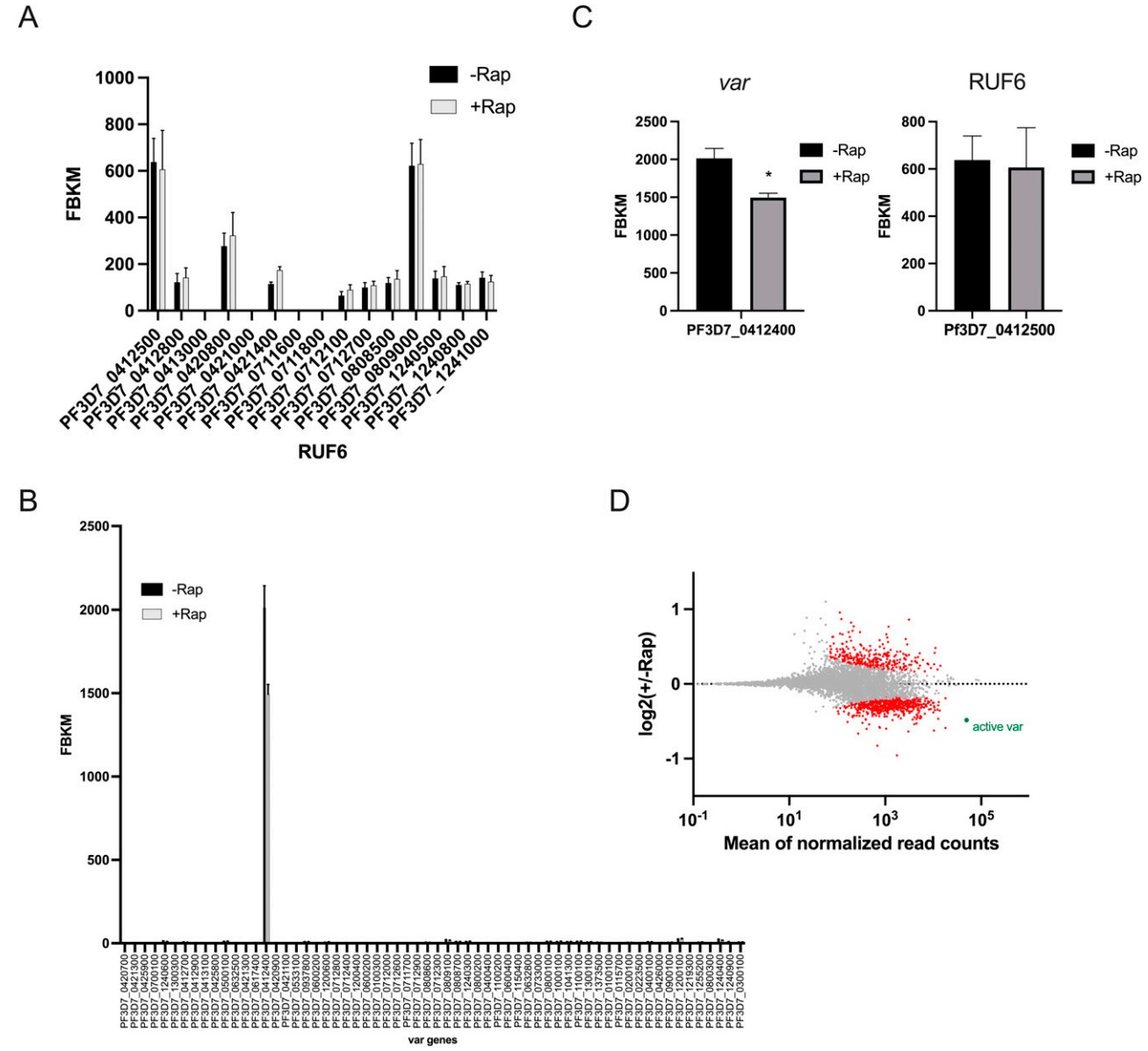

**Figure 6. Pf-DDX5 is involved in the *var* gene transcription.**
**(A, B)** Transcriptional profile of the RUF6 gene family and (B) the *var* gene family at 12 hpi assayed by RNA sequencing for the control –Rap and treated +Rap samples. **(C)** Transcriptional levels of the active *var* gene and the associated RUF6 member at 12 hpi assayed by RNA sequencing for the control –Rap and treated +Rap samples. **(A, D)** MA plot of log$_2$(rapamycin-treated/untreated, M) plotted over the mean abundance of each gene (A) at 12 hpi. Transcripts with a significantly higher (above *x*-axis) or lower (below *x*-axis) abundance in the presence of rapamycin are highlighted in red ($q \leq 0.05$). The active *var* gene is highlighted in green ($q = 2.59 \times 10^{-6}$). Three replicates were used for untreated and rapamycin-treated parasites. *P*-values were calculated with a Wald test for significance of coefficients in a negative binomial generalized linear model as implemented in DESeq2 (Love et al, 2014). $q$ = Bonferroni-corrected *P*-value. Mean_SEM of three independent experiments is shown.

unwind secondary structures of DNA (Wu et al, 2019). Pf-DDX5 was selected for further validation not only because of its canonical role in unwinding RNA/DNA (Wu et al, 2019), but also because of its role in transcriptional regulation and elongation and its interaction directly with RNA Pol II (Clark et al, 2013). Typically, helicases function in the separation of double-stranded RNA, DNA, and RNA/DNA hybrid structures and contribute to the process of gene regulation (Bourgeois et al, 2016). The *P. falciparum* genome has a predicted set of 63 helicases (Reddy et al, 2015), but only very few

have been functionally characterized in malaria parasites (Tuteja, 2017). The RNA helicase DOZI has an important role in the translational repression of mRNA during sexual development (Mair et al, 2006), and the DNA-RecQ helicase is an important chromatin factor associated with multiple roles including DNA replication, genome stability, and heterochromatin organization of clonally variant genes (Claessens et al, 2018; Li et al, 2019).

We obtained genetically modified Pf-DDX5-GFP-tagged transfectants for functional analysis using the knock-sideways system

(Birnbaum et al, 2017). IFA localization studies and Western blot analysis confirmed the nuclear presence of Pf-DDX5 during the ring stage, and RIP RT–qPCR confirmed its association with RUF6 ncRNA (Fig 3). We used GFP-tagged Pf-DDX5 parasites to perform Co-IP assays to explore the interactome of Pf-DDX5. Mass spectrometry (LC-MS/MS) revealed many proteins found in the Pf-DDX5 sample that were also found in the original RUF6 ChIRP-MS. This result validates the ChIRP-MS technique as a method to identify ncRNA interactomes in *P. falciparum* (Fig 4D). Shared proteins had functions related to RNA binding and were involved in processes related to the initiation and regulation of transcription from RNA Pol II, strengthening the notion that Pf-DDX5 is associated with RUF6 ncRNA at the *var* gene RNA Pol II promoter. Of note is that histone variant H2A.Z was significant in the Pf-DDX5 IP with a *P*-value of $1.10 \times 10^{-3}$ and a ratio of 2.73. In the RUF6 ChIRP-MS, however, the *P*-value was not significant ($P = 0.89$). H2A.Z was previously shown to be enriched at active *var* gene promoters (Petter et al, 2011, 2013). The combination of two methods CHIRP-MS and Pf-DDX5 Co-IP-MS provides the first comprehensive catalogue of proteins that interacts with RUF6 ncRNA. It contains numerous proteins with unknown functions.

Because the functional role in gene regulation of Pf-DDX5 in malaria parasites has not been reported, we used the Pf-DDX5 knock-sideways strategy to displace most of the nuclear Pf-DDX5 (Fig 5A). The nuclear depletion results in a significant decrease in the active *var* gene transcription without disrupting monoallelic expression (Fig 6). This result identifies Pf-DDX5 as a novel regulator of transcriptional activation of *var* genes.

What is the possible mechanism of Pf-DDX5 at the *var* promoter? Human DDX5 has recently been reported to be involved in transcriptional regulation resolving secondary structures near RNA Pol II promoters (Wu et al, 2019). One of these structures is G-quadruplexes (G4s). G4s have been found to play a role in various cellular processes including gene transcription (Siddiqui-Jain et al, 2002). A recent report demonstrated that DDX5 can bind not only to RNA G-quadruplexes but also to DNA G-quadruplexes (Wu et al, 2019). In this context, it is important to note that a recent study predicted DNA G4s in *P. falciparum*, using the tool G4Hunter algorithm (Gazanion et al, 2020). The authors showed that the *var* multigene family, specifically, showed a significant enrichment in DNA G4s found in the promoter and first exon (Gazanion et al, 2020). We therefore predict that plasmodial DDX5 helps unwind these structures to allow for proper transcription by RNA Pol II throughout the *var* locus. Notably, an RNA G4-quadruplex is predicted in RUF6. This raises the possibility that this structure recruits Pf-DDX5 to this ncRNA.

Two ribonucleases have been reported to regulate clonally variant RUF6 ncRNA in steady-state RNA levels by degrading the nascent RNA: an exosome-independent PfRNase II (Zhang et al, 2014) and an RNA exosome–associated Rrp6 (Fan et al, 2020). Inactivation of those ribonucleases disrupts heterochromatin-dependent silencing of clonally variant gene families. General activation of the RUF6 gene family in Rrp6 mutant parasites indicated that this ncRNA family is enriched in heterochromatin regions. This observation does support our findings that RUF6 is involved in the activation process of silent var genes in heterochromatin islands.

A recent report developed a catalytically inactive Cas9 system to investigate the chromatin associated with *var* gene loci (promoter and intron) (Bryant et al, 2020). We did find 36 shared proteins between the two studies including CHD1, a chromatin remodeling protein, AP2-domain transcription factor, and ISWI, which was validated in their study to be associated with the active *var* promoter. Three proteins were shared between our RUF6 ChIRP-MS and Pf-DDX5 IP with the proteins found significantly in the *var* promoter: protein with unknown functions Pf3D7_0704300, PHAX domain–containing protein Pf3D7_1021900, and chromatin remodeling protein Pf3D7_1104200. Pf-DDX5 was not significantly found in the study by Bryant and colleagues, and this may be explained by the fact that the isolated chromatin primarily is from silent *var* gene promoters.

In conclusion, here we developed a robust ChIRP-MS technique to identify the first ncRNA–protein interactome that sustains the *P. falciparum* immune evasion strategy. This work opens up new avenues to identify unprecedented regulatory chromatin factors that will shed light on the mechanisms of RUF6 ncRNA recruitment to the *var* gene expression site and singular virulence gene expression in this deadly pathogen. This insight may provide new strategies to target pathogenesis of malaria parasites.

# Materials and Methods

### Parasite culture and synchronization

Asexual blood-stage 3D7 *P. falciparum* parasites were cultured as previously described in Lopez-Rubio et al and W.H.O. (2021). Parasites were cultured in human RBCs (obtained from the Etablissement Français du Sang with approval number HS 2016-24803) in RPMI-1640 culture medium (11875; Thermo Fisher Scientific) supplemented with 10% vol/vol Albumax I (11020; Thermo Fisher Scientific), hypoxanthine (0.1 mM final concentration, C.C.Pro Z-41-M), and 10 mg gentamicin (G1397; Sigma-Aldrich) at 4% hematocrit and under 5% $O_2$ and 3% $CO_2$ at 37°C. Parasite development was monitored by Giemsa staining. Parasites were synchronized by sorbitol (5%, S6021; Sigma-Aldrich) lysis at the ring stage, plasmagel (Plasmion; Fresenius Kabi) enrichment of late stages 24 h later, and an additional sorbitol lysis 6 h after plasmagel enrichment. Parasites were cultured under static conditions with the exception of shaking during the late schizont until an early ring stage. Parasites were harvested at 1–5% parasitemia.

### RNA electrophoretic mobility shift assay (RNA EMSA)

Probe sequences for RUF6 (PF3D7_1241000) and its antisense were amplified with primers containing the SP6 promoter sequence. In vitro transcription was performed on 0.2 µg of the PCR products with the MAXIscript T7/SP6 kit (Ambion) using the SP6 enzyme mix. RNAs were checked for size on a denaturing urea–polyacrylamide gel, and biotin was incorporated on the 3′ end of the RNA fragments using the Pierce RNA 3′ End Biotinylation Kit (Thermo Fisher Scientific). Alternatively, the synthesized 5′ biotinylated RNA probes were also used. RNA EMSAs were performed based on LightShift

Chemiluminescent RNA EMSA Instructions (Thermo Fisher Scientific). RNAs were relaxed and refolded by incubation at 64°C and gradual cooling at 4°C before binding. 20 $\mu$l of binding reaction containing 5 g of nuclear extract; 2 g of tRNA; 10, 20, or 40 fmol of biotinylated probe, and when indicated 8 pmol of unlabeled probe were incubated in REMSA buffer (10 mM Hepes, 20 mM KCl, 1 mM MgCl$_2$, and 1 mM DTT) with RNasin Ribonuclease Inhibitor (Promega) at room temperature for 25 min. After electrophoresis on a native polyacrylamide TBE gel, the transferred RNA was cross-linked to the nylon membrane at 120 mJ/cm$^2$. Detection of biotin-labeled RNA was performed using the Chemiluminescent Nucleic Acid Detection Module (Thermo Fisher Scientific) and imaged with the ChemiDoc XRS+ system (Bio-Rad).

## Chromatin isolation by RNA purification and proteomic mass spectrometry analysis (ChIRP-MS)

ChIRP was performed as previously described (Chu & Chang, 2018) with the following modifications. 4 × 10$^{10}$ parasites per sample were harvested and lysed with saponin before 3% formaldehyde cross-linking. After final wash buffer washes, five additional washes with 500 mM NaCl followed by another five washes with 25 mM NH$_4$HCO$_3$ buffer (ABC) were performed. Bound proteins were analyzed by proteomic mass spectrometry. A total of four biological replicates, cultured separately and grown at different times, were prepared for each of the three samples: scrambled probe, RNase A–treated, and even probe samples.

For proteomic mass spectrometry, bound proteins were first on-bead–digested for 1 h with 0.6 $\mu$g of trypsin–LysC (Promega), then loaded onto a homemade C18 StageTips for desalting. Peptides were eluted using 40/60 MeCN/H$_2$O + 0.1% formic acid and vacuum-concentrated to dryness. Online liquid chromatography (LC) was performed with an RSLCnano system (UltiMate 3000; Thermo Fisher Scientific) coupled to a Q Exactive HF-X with a Nanospray Flex ion source (Thermo Fisher Scientific). Peptides were first trapped on a C18 column (75 $\mu$m inner diameter × 2 cm; nanoViper Acclaim PepMap 100; Thermo Fisher Scientific) with buffer A (2/98 MeCN/H$_2$O in 0.1% formic acid) at a flow rate of 2.5 $\mu$l/min over 4 min. Separation was then performed on a 50 cm × 75 $\mu$m C18 column (nanoViper Acclaim PepMap RSLC, 2 $\mu$m, 100 Å; Thermo Fisher Scientific) regulated to a temperature of 50°C with a linear gradient of 2–30% buffer B (100% MeCN in 0.1% formic acid) at a flow rate of 300 nl/min over 91 min.

MS full scans were performed in the ultrahigh-field Orbitrap mass analyzer in ranges of $m/z$ 375–1,500 with a resolution of 120,000 at $m/z$ 200. The top 20 intense ions were subjected to Orbitrap for further fragmentation via high-energy collision dissociation activation and a resolution of 15,000 with the intensity threshold kept at 1.3 × 10$^5$. We selected ions with a charge state from 2+ to 6+ for screening. Normalized collision energy was set at 27, and the dynamic exclusion, at 40 s.

For identification, the data were searched against the Plasmodium falciparum FASTA database (*PlasmoDB-36 Pfaciparum3D7 AnnotatedProtein containing cas9 and the common contaminants*) using Sequest$^{HT}$ through Proteome Discoverer (version 2.2 or 2.4). Enzyme specificity was set to trypsin, and a maximum of two-missed cleavage sites were allowed. Oxidized methionine, carbamidomethyl-cysteine, and N-terminal acetylation were set as variable modifications. The

maximum allowed mass deviation was set to 10 ppm for mono-isotopic precursor ions and 0.02 D for MS/MS peaks. The resulting files were further processed using myProMS (Poullet et al, 2007), v3.9.3 (https://github.com/bioinfo-pf-curie/myproms). FDR calculation used Percolator (The et al, 2016) and was set to 1% at the peptide level for the whole study. The label-free quantification was performed by peptide extracted ion chromatograms (XICs) computed with MassChroQ, version 2.2.1 (Valot et al, 2011). For protein quantification, XICs from proteotypic peptides shared between compared conditions (TopN matching) with two-missed cleavages were used. Median and scale normalization was applied to the total signal to correct the XICs for each biological replicate (n = 4). To estimate the significance of the change in protein abundance, a linear model (adjusted on peptides and biological replicates) was performed and $P$-values were adjusted with a Benjamini–Hochberg FDR procedure. Proteins with at least twofold increase between probe and control condition, an adjusted $P$-value less than 0.05, and at least three total peptides in all replicates were considered significantly enriched in sample comparisons. Candidate proteins that display at least three total peptides in all replicates were chosen if they were unique in the target sample, even probes compared with at least one of the controls.

## RNA isolation and reverse transcription–quantitative PCR (RT–qPCR)

RNA was harvested from synchronized parasite cultures after saponin lysis in 0.075% saponin in PBS, followed by one wash in PBS and resuspension in the QIAzol reagent. Total RNA was extracted using an miRNeasy minikit and by performing on-column DNase treatment (217004; Qiagen). Reverse transcription from ChIRP eluted RNA was achieved using SuperScript VILO (Thermo Fisher Scientific) and random hexamer primers. cDNA levels were quantified by quantitative PCR in the CFX384 real-time PCR detection system (Bio-Rad) using Power SYBR Green PCR Master Mix (Applied Biosystems) and primers from a previous study (Guizetti et al, 2016). Starting quantity means of three replicates were extrapolated from a standard curve of serial dilutions of genomic DNA. RUF6 and housekeeping fructose–bisphosphate aldolase (PF3D7_1444800) transcript levels were compared in ChIRP and input samples. The starting quantity means of three replicates were extrapolated from a standard curve of serial dilutions of genomic DNA.

## Generation of genetically modified *P. falciparum* strains

All cloning of episomal constructs was performed using KAPA HiFi DNA Polymerase (07958846001; Roche), In-Fusion HD Cloning Kit (639649; Clontech), and XL10-Gold Ultracompetent *E. coli* (200315; Agilent Technologies). Transgenic GFP-tagged parasites were generated as previously described in Birnbaum et al (2017). For localization and knock-sideways, the last 500–1,000 bp of each candidate protein target gene was cloned into pSLI-2×FKBP-GFP. Each sequence started with an in-frame stop codon, but the stop codon at the end of the gene was removed. 50 $\mu$g of plasmid DNA was transfected into ring-stage 3D7 *P. falciparum* parasites using the protocol described elsewhere (Hasenkamp et al, 2013). Transfected parasites were selected with constant drug selection

pressure of 4 nM WR99210 (Jacobus Pharmaceuticals) to obtain a cell line containing the episomal plasmid. A second drug selection using 400 µg/ml of G418 was done to select for integrants. Once parasites emerged, gDNA of each integration cell line was collected using a commercial kit (DNeasy Blood & Tissue Kit) and checked by PCR to show that integration occurred at the correct locus. Both genome- and vector-specific primers for the 5′ and 3′ regions were used so that the PCR product would cover the plasmid/genome junction. Vector primers used were the same as described in Birnbaum et al (2017). Once proper size gel bands from PCR were seen, localization of the GFP-tagged protein was checked with a fluorescence microscope (Delta Vision Elite microscope; GE Healthcare). Image overlays were generated using Fiji (Schindelin et al, 2012).

## Procedure for knock-sideways

pSLI-2×FKBP-GFP-tagged parasites were further transfected with 50 µg of mislocalizer plasmid DNA. The PPM mislocalizer (pLyn-FRB-mCherry) for nuclear targets was used. Parasites were selected with 2 µg/ml blasticidin S. Once transfectants were obtained, mislocalizer mCherry expression was assessed using a fluorescence microscope. For knock-sideways experiments, 20 µl rapalog working solution (resulting in 250 nM final concentration) was added to cause mislocalization of pSLI-2×FKBP-GFP-tagged parasites to the PPM. Parasites were cloned by limiting dilution, and the targeted genomic locus was sequenced to confirm tag and FKBP integration, and epitope mislocalizer BSD.

## Western blot analysis

iRBCs were washed once with Dulbecco's phosphate-buffered saline (DPBS, 14190; Thermo Fisher Scientific) at 37°C and lysed with 0.075% saponin (S7900; Sigma-Aldrich) in DPBS at 37°C. Parasites were washed once with DPBS, resuspended in 1 ml cytoplasmic lysis buffer (25 mM Tris–HCl, pH 7.5, 10 mM NaCl, 1.5 mM MgCl$_2$, 1% IGEPAL CA-630, and 1× protease inhibitor cocktail ["PI," 11836170001; Roche]) at 4°C, and incubated on ice for 30 min. Cells were further homogenized with a chilled glass Dounce homogenizer, and the cytoplasmic lysate was cleared with centrifugation (13,500$g$, 10 min, 4°C). The pellet (containing the nuclei) was resuspended in 100 µl nuclear extraction buffer (25 mM Tris–HCl, pH 7.5, 600 mM NaCl, 1.5 mM MgCl$_2$, 1% IGEPAL CA-630, and PI) at 4°C and sonicated for 10 cycles with 30-s (on/off) intervals (5-min total sonication time) in Diagenode Pico Bioruptor at 4°C. This nuclear lysate was cleared with centrifugation (13,500$g$, 10 min, 4°C). Protein samples were supplemented with NuPAGE sample buffer (NP0008; Thermo Fisher Scientific) and NuPAGE reducing agent (NP0004; Thermo Fisher Scientific) and denatured for 5 min at 95°C. Proteins were separated on a 4–15% TGX (Tris-Glycine eXtended) (Bio-Rad) and transferred to a PVDF membrane. The membrane was blocked for 1 h with 5% milk in PBST (PBS and 0.1% Tween-20) at 25°C. GFP-tagged proteins and histone H3 were detected with anti-GFP (1:1,000 in 5% milk–PBST; ChromoTek) and anti-H3 (ab1791, 1:1,000 in 5% milk–PBST; Abcam) primary antibodies, respectively, followed by donkey anti-rabbit secondary antibody conjugated to HRP (GENA934, 1:5,000 in 5% milk–PBST; Sigma-Aldrich). Aldolase was

detected with anti-aldolase-HRP (ab38905, 1:5,000 in 5% milk–PBST; Abcam). HRP signal was developed with a SuperSignal West Pico chemiluminescent substrate (34080; Thermo Fisher Scientific) and imaged with a ChemiDoc XRS+ system (Bio-Rad).

## Immunofluorescence assay

For live-cell fluorescence microscopy, 500 µl of parasite cultures (3–5% parasitemia and 3–4% hematocrit) was spun down for 45 s at 400$g$. The iRBC pellet was resuspended in PBS-DAPI for 20 min before spinning down for 45 s at 2.4 and removing the supernatant. The iRBC pellet was then washed with PBS before mounting. Unattached cells were washed out with PBS and finally covered with a culture medium prepared with phenol red–free RPMI 1640. Images were captured using a Delta Vision Elite microscope (GE Healthcare). Image overlays were generated using Fiji (Schindelin et al, 2012).

SLI-transfected cultures were used with GFP-booster antibodies, anti-GFP V$_H$H/nanobody conjugated to a fluorescent dye (ChromoTek) and a ChromoTek RFP antibody (6G6) to visualize mCherry mislocalizer. 10 µl of iRBCs was washed with PBS and fixed with 0.0075% glutaraldehyde/4% PFA/PBS for 30 min. After PBS washing, parasites were permeabilized with 0.1% Triton X-100/PBS for 10–15 min before quenching free aldehyde groups with NaBH$_4$ solution for 10 min. Next, parasites were blocked with 3% BSA–PBS for 30 min. Primary antibody incubation lasted for 1 h before three washes with PBS, and secondary antibody incubation, for 30–60 min; Alexa Fluor 488–conjugated anti-mouse IgG (Invitrogen) was diluted 1:2,000 in 4% BSA–PBS. After three final washes in PBS, the cells were mounted in Vectashield containing DAPI for nuclear staining. Images were captured using a Delta Vision Elite microscope as described before.

## RNA immunoprecipitation RT–qPCR

WT 3D7, pSLI-2×FKBP-GFP-DDX5 (Pf3D7_1445900)–tagged parasites, and pSLI-2×FKBP-GFP protein with unknown function (Pf3D7_1423700)–tagged parasites were synchronized and harvested at 18 hpi (n = 3 biological replicates). Each culture (10$^9$ parasites) was centrifuged, and RBCs were lysed with six volumes of 0.15% saponin in DPBS for 5 min at 4°C. Parasites were centrifuged at 4,000$g$ for 5 min at 4°C, and the pellet was washed twice with DPBS at 4°C. Parasites were then cross-linked with 3% formaldehyde for 15 min at room temperature and quenched with 125 mM glycine for 5 min on ice. Each sample was resuspended in 2 ml lysis buffer (10 mM Hepes, pH 8, 10 mM KCl, 0.1 mM EDTA, pH 8, and 0.25% final concentration of IGEPAL CA-630) + CI (EDTA free; Roche) at 4°C and incubated with gentle agitation at 4°C for 30 min. Extracts were centrifuged in microcentrifuge tubes for 10 min at 13,500$g$ at 4°C. Once the supernatant was removed, the pellet was resuspended in 300 µl SDS lysis buffer (50 mM Tris–HCl, pH 8, 10 mM EDTA, pH 8, and 1% SDS + CI) at 4°C. Next, 300 µl was transferred to Diagenode 1.5-ml sonication tubes. Samples were sonicated with the Diagenode Pico sonicator for 22 cycles (30 s on/30 s off), vortexing and spinning occasionally. Lysates were centrifuged at 16,000 × $rcf$ for 10 min at 4°C, and the supernatants were transferred to fresh tubes. Beads were prepared by washing 25 µl per sample with 500 µl of ice-cold

dilution buffer. Samples then had 2× volume of dilution buffer (10 mM Tris, pH 7.5, 150 mM NaCl, and 0.5 mM EDTA) + CI (EDTA free; Roche) added and washed beads (gtma-10 GFP-Trap Magnetic Agarose beads; ChromoTek) and were incubated overnight at 4°C. The next morning, beads were collected and washed with 500 µl of wash buffer (2×SSC, 0.5% SDS, and 1 mM PMSF) for 5 min with rotation at 4°C. After the final wash, beads were resuspended in 100 µl pK buffer. Input sample volume was adjusted to 100 µl with pK buffer (100 mM NaCl, 10 mM Tris–HCl, pH 7.0, 1 mM EDTA, 0.5% SDS, and 5% pK [proteinase K, P8107S; New England Biolabs]). Samples were incubated at 50°C for 45 min with constant mixing. Next, samples were boiled at 95°C for 10 min before being resuspended in 700 µl of QIAzol reagent (79306; Qiagen). RNA was extracted using an miRNeasy minikit and by performing on-column DNase treatment (217004; Qiagen). Reverse transcription from RNA-IP eluted RNA was achieved using SuperScript VILO (Thermo Fisher Scientific) and random hexamer primers. cDNA levels were quantified by quantitative PCR in the CFX384 real-time PCR detection system (Bio-Rad) using Power SYBR Green PCR Master Mix (Applied Biosystems) and primers from a previous study (Guizetti et al, 2016). The starting quantity means of three replicates were extrapolated from a standard curve of serial dilutions of genomic DNA. RUF6, housekeeping fructose–bisphosphate aldolase (PF3D7_1444800), and alanine tRNA (PF3D7_0411500) transcript levels were compared in RNA IP and input samples.

## Co-immunoprecipitation followed by mass spectrometry (Co-IP-MS)

pSLI-2×FKBP-GFP-DDX5-tagged parasites (n = 5 biological replicates) and pSLI-2×FKBP-GFP protein with unknown function–tagged parasites (n = 5 biological replicates) were synchronized. At 18 hpi, each culture (1.5 × 10$^9$ parasites) was centrifuged and RBCs were lysed with six volumes of 0.15% saponin in DPBS for 5 min at 4°C. Parasites were centrifuged at 4,000g for 5 min at 4°C, and the pellet was washed twice with DPBS at 4°C. Parasites were then crosslinked with 3% formaldehyde for 15 min at room temperature and quenched with 125 mM glycine for 5 min on ice. Cross-linked parasites were washed twice with DPBS and then lysed with 5 ml of lysis buffer (10 mM Tris–HCl, pH 7.5, 1 mM EDTA, and 0.5% IGEPAL CA-630) supplemented with protease inhibitors (78440; Thermo Fisher Scientific) and 1 U/µl of Benzonase (71206; Merck) at 4°C and incubated with rotation for 30 min at 4°C. Extracts were centrifuged for 8 min at 380g at 4°C, and the cytoplasmic supernatant was removed. The nuclear pellet was resuspended in 900 µl nuclear lysis buffer (10 mM Tris–HCl, pH 7.5, 150 mM NaCl, 1 mM EDTA, 0.1% sodium deoxycholate, 0.1% SDS, and PI) at 4°C and transferred to 1.5-ml sonication tubes (300 µl per tube, C30010016; Diagenode). Samples were sonicated for 10 min (30 s on/off) in Diagenode Pico Bioruptor at 4°C. Lysates were then centrifuged for 10 min at 13,500g at 4°C, and the supernatant was transferred to a fresh tube. 900 µl of the supernatant was mixed with 1.35 ml (2:3) dilution buffer (10 mM Tris–HCl, pH 7.5, 150 mM NaCl, and 0.5 mmM EDTA). Nuclear fraction supernatants were incubated with 25 µl of α-GFP trap beads (ChromoTek), first washed twice in dilution buffer, overnight with rotation at 4°C. The next day, the beads were collected on a magnet and the supernatant was removed. While on the magnetic

stand, beads were washed twice with 500 µl washing buffer (10 mM Tris–HCl, pH 7.5, 150 mM NaCl, 0.5 mM EDTA, and 0.05% NP-40) and once with 25 mM NH$_4$HCO$_3$ (09830; Sigma-Aldrich) buffer, and then transferred to a new tube. Finally, the beads were resuspended in 100 µl of 25 mM NH$_4$HCO$_3$ (09830; Sigma-Aldrich) and digested by adding 0.2 µg of trypsin–LysC (Promega) for 1 h at 37°C. Samples were then loaded into custom-made C18 StageTips packed by stacking one AttractSPE Disk (#SPE-Disks-Bio-C18-100.47.20; Affinisep) and 2 mg beads (#186004521, Sep-Pak C18 Cartridge; Waters) into a 200-µl micropipette tip for desalting. Peptides were eluted using a ratio of 40:60 CH$_3$CN:H$_2$O + 0.1% formic acid and vacuum-concentrated to dryness with a SpeedVac apparatus. Peptides were reconstituted in 10 of injection buffer in 0.3% TFA before liquid chromatography–tandem mass spectrometry (LC-MS/MS) analysis. Online LC was performed as described previously in the ChIRP-MS protocol and coupled to an Orbitrap Exploris 480 mass spectrometer (Thermo Fisher Scientific) by modifying the peptide trapping flow to 3.0 µl/min over 4 min and the separation temperature to 40°C with a linear gradient of 3–29% buffer B.

## Parasite growth assay

Parasite growth was measured as described previously (Vembar et al, 2015). A clone of pSLI-DDX5-GFP-FKBP with mCherry-FRB was tightly synchronized and diluted to 0.2% parasitemia (5% hematocrit) at the ring stage using the blood of two different donors separately. Each culture was split, and 20 µl rapalog working solution was added (250 nM final concentration of rapamycin) to one half. The growth curve was performed with three technical replicates per condition per blood. Parasitemia was measured every 24 h by counting 10 randomly selected different fields on Giemsa-stained slides each day for a total of 5 d.

## Stranded RNA sequencing and analysis

Three biological replicates, grown and harvested at separate times, were used for RNA sequencing. Parasites were synchronized by sorbitol (5%, S6021; Sigma-Aldrich) lysis at the ring stage, plasmagel (Plasmion; Fresenius Kabi) enrichment of late stages 24 h later, and an additional sorbitol lysis 3 h after plasmagel enrichment before the parasites were separated into the ± rapamycin groups. After a full cycle, another sorbitol was done (at 3 hpi) before harvesting at 12 hpi (9 h later and with 60 h of rapamycin exposure to the + rapamycin group). The ± rapamycin-infected RBCs containing synchronized (12 hpi ± 3 h) parasites were lysed in 0.075% saponin (S7900; Sigma-Aldrich) in DPBS at 37°C. The parasite cell pellet was washed once with DPBS and then resuspended in 700 µl QIAzol reagent (79306; Qiagen). Total RNA was subjected to rRNA depletion to ensure ncRNA and mRNA capture using the RiboCop rRNA Depletion Kit (Lexogen) before strand-specific RNA-seq library preparation using TruSeq Stranded RNA LT Kit (Illumina) with the KAPA HiFi polymerase (Kapa Biosystems) for the PCR amplification. Multiplexed libraries were subjected to 150-bp paired-end sequencing on a NextSeq 500 platform (Illumina). Sequenced reads (150-bp paired-end) were mapped to the *P. falciparum* genome (Gardner et al, 2002) (plasmoDB.org; version 3, release 57) using "bwa mem" (Li & Durbin, 2009) allowing a read to align only once to

the reference genome (option "−c 1"). Alignments were subsequently filtered for duplicates and reads with a mapping quality ≥ 20 using SAMtools (Li & Durbin, 2009). Three biological replicates for −Rap and +Rap samples were analyzed for the knockdown experiment.

### Estimation of cell cycle progression

To estimate the parasite age (i.e., hours post-infection), we correlated transcript levels (average FPKM ≥ 10) from all three replicates of the +rapamycin and −rapamycin samples with the transcript levels of a reference microarray dataset (Bozdech et al, 2003). Pearson's r correlation coefficients were calculated and visualized in R () with options cor (R Core Team, 2020) and heatmap2(), respectively.

### Differential gene expression analysis

A clone of integrated pSLI-DDX5-GFP-FKBP with episomal mCherry-FRB was synchronized and split into two cultures. Rapamycin (250 nM final concentration) was added to one culture at 0 hpi, and parasites were harvested 60 h later during the ring stage (12 hpi). RNA-sequencing reads for three technical replicates of the rapamycin-treated and three technical replicates of the untreated pSLI-DDX5-GFP-FKBP with mCherry-FRB clone were mapped to the *P. falciparum* genome and quality-filtered as described above for RNA-seq. Strand-specific gene counts were calculated using htseq-count (Anders et al, 2015). Differential gene expression analysis was performed using DESeq2 (Love et al, 2014) with significantly differentially expressed genes featuring a Benjamini–Hochberg-adjusted *P*-value (i.e., *q*) ≤ 0.05. MA plots were generated using the "baseMean" (mean normalized read count over all replicates and conditions) and "log$_2$FoldChange" values (rapamycin-treated over control) as determined by DESeq2. RPKM values were calculated in R using the command rpkm() from the package edgeR (Robinson et al, 2010). Gene Ontology enrichments were calculated using the built-in tool at PlasmoDB.org (Aurrecoechea et al, 2017).

### Statistical analysis

All statistical analyses were performed using GraphPad Prism, version 9.1.0 (216), for Mac. To test for a normal distribution of the data, the Shapiro–Wilk normality test was used. To test for significance between the two groups, a two-sided independent-samples *t* test was used. Gene Ontology enrichments were calculated using the build-in tool at https://plasmoDB.org

## Data Availability

The mass spectrometry proteomics data have been deposited to the ProteomeXchange consortium via the PRIDE partner repository with the dataset identifier PXD036801. The data generated in this study are available in the following databases: NCBI for the RNA-seq: BioProject accession number PRJNA875234.

## Supplementary Information

## Acknowledgements

We would like to thank Jessica Bryant for helpful comments and critically reading the article. This work was supported by ERC grants to A Scherf (ERC AdG PlasmoSilencing) and S Baumgarten (ERC StG PlasmoEpiRNA) and the Agence Nationale de Recherche (grant ANR-11 LabEx-0024-01 ParaFrap to A Scherf).

### Author Contributions

GM Diffendall: conceptualization, investigation, and writing—original draft.
A Barcons-Simon: investigation and methodology.
S Baumgarten: investigation and methodology.
F Dingli: investigation and methodology.
D Loew: investigation and methodology.
A Scherf: conceptualization and writing—original draft.

### Conflict of Interest Statement

The authors declare that they have no conflict of interest.

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
