## [Reviewer comments · Life Science Alliance]

Life Science Alliance

Discovery of RUF6 ncRNA interacting proteins involved in *P. falciparum* immune evasion

Gretchen Diffendall, Anna Barcons-Simon, Sebastian Baumgarten, Florent Dingli, Damarys Loew, and Artur Scherf
DOI: <https://doi.org/10.26508/lsa.202201577>

Corresponding author(s): Artur Scherf, Institut Pasteur

Review Timeline:

Submission Date:	2022-06-24
Editorial Decision:	2022-08-05
Revision Received:	2022-09-07
Editorial Decision:	2022-10-11
Revision Received:	2022-10-25
Accepted:	2022-10-26

Transaction Report:

August 5, 2022

Re: Life Science Alliance manuscript #LSA-2022-01577-T

Prof. Artur Scherf
Institut Pasteur
25, rue du Dr. Roux
Paris 75724
France

Dear Dr. Scherf,

Thank you for submitting your manuscript entitled "Discovery of RUF6 ncRNA interacting proteins involved in *P. falciparum* immune evasion" to Life Science Alliance. The manuscript was assessed by expert reviewers, whose comments are appended to this letter. We invite you to submit a revised manuscript addressing the Reviewer comments.

Thank you for this interesting contribution to Life Science Alliance. We are looking forward to receiving your revised manuscript.

Sincerely,

- A letter addressing the reviewers' comments point by point.
- An editable version of the final text (.DOC or .DOCX) is needed for copyediting (no PDFs). Line numbers would be helpful.
- High-resolution figure, supplementary figure and video files uploaded as individual files: See our detailed guidelines for preparing your production-ready images, <https://www.life-science-alliance.org/authors>
- Summary blurb (enter in submission system): A short text summarizing in a single sentence the study (max. 200 characters including spaces). This text is used in conjunction with the titles of papers, hence should be informative and complementary to the title and running title. It should describe the context and significance of the findings for a general readership; it should be written in the present tense and refer to the work in the third person. Author names should not be mentioned.
- By submitting a revision, you attest that you are aware of our payment policies found here: <https://www.life-science-alliance.org/copyright-license-fee>

B. MANUSCRIPT ORGANIZATION AND FORMATTING:

Reviewer #1 (Comments to the Authors (Required)):

Diffendall et al developed a novel method, ChIRP-MS, to identify proteins interacting with non-coding RNA, in this case RUF6. Of the multiple proteins identified, they focused the rest of the manuscript on the helicase PfDDX5. They immunoprecipitated PfDDX5 and validated some of the previous interacting proteins. Finally, they performed a 'knock sideways approach' which resulted in PfDDX5 levels being decreased in the nucleus, and a limited decrease of var gene transcript level. Of interest, DDX5 is predicted to interact with DNA G-quadruplexes, which are common within and upstream of var genes.

Overall, the methods and data presented here are entirely novel, and the claims made here are convincing. The manuscript is of great interest to the 'var gene community', but more broadly it opens a new door of investigation on transcription regulation. I only have minor recommendations.

Disclaimer: I don't have hands-on experience with Mass Spec, I can't comment on that method.

Results:

- What is the rationale for performing ChIRP at 18 hours post-invasion?

- Fig 6B/C : The downregulation of var gene expression is modest. Although the parasite populations are said to be synchronised, there is a risk that the Rap+ population is younger by a few hours than the Rap- population, that would result in the difference of var gene expression observed here. To rule out this is the case, the developmental age of each parasite population can be relatively easily determined by correlating the available transcriptome against a known timecourse .

- Some confusion in the main text because Fig6D is mentioned before Fig 6A.

- In the section 'Analysis of ChIRP-MS interacting proteins':

3) "most malaria species" should be "Plasmodium species"

Text refers to 16 candidate proteins but Fig 2D table only shows 15.

- In the Discussion, at the end of the 4th paragraph: Claessens2018 (PlosGen) is a more relevant reference than Claessens2014.

Methods

- Were parasites grown under static conditions or on a shaker?

- In Methods, please indicate the probe sequences used for RUF6. It says against Pf3D7_1241000, but are the probes picking up all RUF? Because in the main text it is reported that the active RUF6 is PF3D7_0412800.

- Chirp-MS: last sentence of 1st paragraph: specify the four samples (as indicated in Results). Please indicate at what step exactly the four biological replicates are generated. Same for Fig6D, indicate what is meant by replicate.

- Methods for RNAseq is not mentioned. I don't know the Journal policy but in my opinion the data should be made publicly available.

- I can't find Table S1?

Reviewer #2 (Comments to the Authors (Required)):

In the manuscript by Diffendall et al. the authors set out to identify proteins interacting with RUF6 non-coding RNAs, which is involved in the transcriptional regulation of the var gene family encoding the major virulence protein family of the malaria parasite *P. falciparum*. To this end they established a ChIRP-MS protocol for the first time in *P. falciparum*, which involves immunoprecipitation of proteins bound to chromatin-associated RNAs, followed by their identification by mass-spectrometry. One of the identified candidate proteins is the helicase Pf-DDX5, which the authors further characterize functionally by knock-sideways (KS) in conjunction with RNAseq, as well as by quantitative label-free mass spectrometry. While the ChIRP-MS data appear robust and the technique represents a significant methodological advance, the analysis of the transgenic Pf-DDX5-GFP and knock-sideways parasite lines is less convincing and requires further controls to substantiate the conclusion that Pf-DDX5 functions in complex with RUF6 in var gene regulation.

Specific comments:

- The parasite lines generated in this study should be characterized in more detail. What RUF6 and var genes are dominantly expressed in the two SLI lines? RUF6 is a multigene family, so do the used primers amplify all members equally? Could the lack of RUF6 enrichment by RIP in the transgenic Pf3D7_1423700-GFP parasites be due to lower/alternative transcription/detectability of certain RUF6 variants and consequently var?
- The choice of Pf3D7_1423700 as a control for identifying Pf-DDX5 interacting proteins in quantitative MS is questionable, since both were originally identified by ChIRP-MS in complex with RUF6 ncRNA. So even if Pf3D7_1423700 does not directly bind to RUF6 RNA, there would be the possibility that it interacts with Pf-DDX5 or shares part of the complex. Therefore, a different control should be included to validate these results (for example GFP trap of 3D7 parasites or better parasites expressing GFP alone).
- PfDDX5 is mostly cytoplasmic rather than nuclear in rings and also schizonts according to Fig. 3B. Please comment.
- Table S12 should represent GO analysis of the 26 genes shared between DDX5 MS and RUF6 ChIRP listed in table S10. However, there are several genes popping up in the GO list which are not present in Table S10 (e.g. PF3D7_0320900, PF3D7_0629200).
- The knock-sideways data are not entirely convincing. What is the phenotype of the parasites 5 days post induction? Is the "fitness cost" due to a shift in developmental cycle or to less invasion? How does the parental parasite line grow after 5 days rapamycin treatment? The phenotype needs to be established and controlled in more detail. What is the explanation for reduction in total PfDDX5 level in Fig 5B? Please also show the cytoplasmic fraction.
- Rapamycin treatment of Pf-DDX5 for 60 hours resulted in transcriptional changes in 869 genes (app. 17% of the genome), and GO analysis would be consistent with a shift in the developmental age of the parasites (e.g. upregulation of schizont specific processes such as invasion, downregulation of ring specific processes such as translation and RNA processing). This may also affect the relative abundance of var gene transcripts. What are the transcriptional changes in the parental line after 60 hours rapamycin treatment?
- Suppl. Figure S2: the few rings shown in a single field of view of the slides may not be representative, so a more detailed breakdown of stage composition would be critical since the transcriptional changes point to upregulation of invasion related genes which would be consistent with more residual schizonts remaining in the RAPA treated culture due to a gross growth delay. This would affect the relative level of var gene transcripts too. Ideally, var transcription should be monitored over several consecutive time points in the life cycle.
- Is there a difference in PfEMP1 expression after Pf-DDX5 KS? This could for example be addressed by western blot with anti-ATS antibodies.
- The active var gene was the only deregulated var gene. What about other VSAs? According to Barcon-Simmons et al, other heterochromatic families such as PfMC2TM were also deregulated by ruf6 KD. Do you observe this as well with PfDDX5 KS?

Minor comments:

- Introduction page 3 paragraph 1: full stop missing after citation.
- Results, paragraph 1: change "...and was reproducible" to "...that was reproducible".
- Table S1 is missing???
- Please specify which selection criteria apply to which candidate in Fig. 2D.
- Please provide data on validation of the transgenic parasite lines generated in this study, for example the PCR screen mentioned in the M&M section showing correct integration and lack of wt locus.
- Materials and Methods: EMSA: in vitro transcription on 0.2 g of the PCR product is probably incorrect. Please provide sequences for ruf6 probes.
- Materials and Methods: RNA IP: please specify "dilution buffer", please specify what "beads" were used.
- Materials and Methods: RNAseq and differential gene expression analysis: how were the parasites synchronized to 0 hpi? And were the 60h rapa treated parasites indeed sorbitol synchronized to 12 hpi +/- 3h again before the saponin lysis, or does this refer to synchronization at the start of rapamycin treatment?
- Materials and Methods: Co-IP-MS of GFP tagged proteins is referred to as "quantitative MS" in the text, but there seems no labelling involved. Please provide some information on normalization and data analysis, also see comments before regarding controls for this experiment.

Point by point reply to reviewer's comments;**Reviewer #1**

Diffendall et al developed a novel method, ChIRP-MS, to identify proteins interacting with non-coding RNA, in this case RUF6. Of the multiple proteins identified, they focused the rest of the manuscript on the helicase PfDDX5. They immunoprecipitated PfDDX5 and validated some of the previous interacting proteins. Finally, they performed a 'knock sideways approach' which resulted in PfDDX5 levels being decreased in the nucleus, and a limited decrease of var gene transcript level. Of interest, DDX5 is predicted to interact with DNA G-quadruplexes, which are common within and upstream of var genes.

Overall, the methods and data presented here are entirely novel, and the claims made here are convincing. The manuscript is of great interest to the 'var gene community', but more broadly it opens a new door of investigation on transcription regulation. I only have minor recommendations. Disclaimer: I don't have hands-on experience with Mass Spec, I can't comment on that method.

Results:

1.- What is the rationale for performing ChIRP at 18 hours post-invasion?

Reply: We chose this timepoint because both RUF6 and var genes are transcribed at this time point (see lines 200-202).

2- Fig 6B/C : The downregulation of var gene expression is modest. Although the parasite populations are said to be synchronised, there is a risk that the Rap+ population is younger by a few hours than the Rap- population, that would result in the difference of var gene expression observed here. To rule out this is the case, the developmental age of each parasite population can be relatively easily determined by correlating the available transcriptome against a known timecourse.

Reply: We had addressed the synchronization between the Rap+ and Rap- populations before starting the RNA-seq analysis. The bioinformatic analysis shows no growth difference between both populations and indicates that most parasites are between 8-9 hpi. We have added the bioinformatic analysis as supplementary figure 3 to the manuscript (see lines 310-312 and 737-742).

Supplementary Figure 3

Figure Legend

Heatmap of Pearson r correlation coefficient calculated between transcript levels in +/-rapamycin samples and transcript levels of a reference transcriptome dataset (Bozdech et al., 2003)

Bozdech Z, Llinás M, Pulliam BL, Wong ED, Zhu J, DeRisi JL. The transcriptome of the intraerythrocytic developmental cycle of *Plasmodium falciparum*. PLoS Biol. 2003 Oct;1(1):E5.

Materials & Methods

To estimate the parasite age (i.e. hours post infection), we correlated transcripts levels (average FPKM ≥ 10) from all three replicates of the +rapamycin and -rapamycin samples with the transcript levels of a reference microarray dataset (Bozdech et al., 2003). Pearson r correlation coefficients were calculated and visualized in R () with options cor (R Core Team (2020)) and heatmap2(), respectively.

R Core Team (2020). R: A language and environment for statistical computing. R Foundation for Statistical Computing, Vienna, Austria. URL: <https://www.R-project.org/>.

3- Some confusion in the main text because Fig6D is mentioned before Fig 6A.

Reply: We agree with the reviewer and have re-worded the text to eliminate any confusion (see lines 312-319).

4- In the section 'Analysis of ChIRP-MS interacting proteins':

3) "most malaria species" should be "Plasmodium species"

Text refers to 16 candidate proteins but Fig 2D table only shows 15.

Reply: We thank this reviewer for pointing out those inaccuracies in the text. We have corrected them in the manuscript (see lines 231 and 232).

5- In the Discussion, at the end of the 4th paragraph: Claessens2018 (PlosGen) is a more relevant reference than Claessens 2014.

Reply: We have changed the reference as recommended by this reviewer (see lines 398 and 852-855).

Methods

6- Were parasites grown under static conditions or on a shaker?

Reply: Parasites were grown under static conditions and only put on the shaker for re-invasion. We have added this information to the Methods section (see lines 474-476).

7- In Methods, please indicate the probe sequences used for RUF6. It says against Pf3D7_1241000, but are the probes picking up all RUF? Because in the main text it is reported that the active RUF6 is PF3D7_0412800.

Reply: Because RUF6 members are highly homologous, the probes used in this work will bind to all other members.

8- Chirp-MS: last sentence of 1st paragraph: specify the four samples (as indicated in Results). Please

indicate at what step exactly the four biological replicates are generated. Same for Fig6D, indicate what is meant by replicate.

Reply: We have specified the samples (see lines 502-504). The parasite samples were cultured separately and grown at a different time. This has been clarified in the manuscript: Methods section (see lines 502-504 and 717-718).

9- Methods for RNAseq is not mentioned. I don't know the Journal policy but in my opinion the data should be made publicly available.

Reply: We thank the reviewer for pointing this out. The data is available on the following databases (see lines 767-769):

PRIDE for the MS from Institut Curie (accession number will be available within a week)
NCBI for the RNA-seq: BioProject accession number PRJNA875234

10- I can't find Table S1?

Reply: We excuse for having missed to add table S1 to the manuscript submission. It has been added to the revised version of the manuscript (see line 1115).

Reviewer #2

In the manuscript by Diffendall et al. the authors set out to identify proteins interacting with RUF6 non-coding RNAs, which is involved in the transcriptional regulation of the var gene family encoding the major virulence protein family of the malaria parasite *P. falciparum*. To this end they established a ChIRP-MS protocol for the first time in *P. falciparum*, which involves immunoprecipitation of proteins bound to chromatin-associated RNAs, followed by their identification by mass-spectrometry. One of the identified candidate proteins is the helicase Pf-DDX5, which the authors further characterize functionally by knock-sideways (KS) in conjunction with RNAseq, as well as by quantitative label-free mass spectrometry. While the ChIRP-MS data appear robust and the technique represents a significant methodological advance, the analysis of the transgenic Pf-DDX5-GFP and knock-sideways parasite lines is less convincing and requires further controls to substantiate the conclusion that Pf-DDX5 functions in complex with RUF6 in var gene regulation.

Specific comments:

1.- The parasite lines generated in this study should be characterized in more detail. What RUF6 and var genes are dominantly expressed in the two SLI lines? RUF6 is a multigene family, so do the used primers amplify all members equally? Could the lack of RUF6 enrichment by RIP in the transgenic Pf3D7_1423700-GFP parasites be due to lower/alternative transcription/detectability of certain RUF6 variants and consequently var?

Reply: The q-PCR primers for RUF6 will amplify all members equally due to the highly conserved sequence between RUF6 members. Since the PCR primer detects all members, we expect no bias for our RIP experiment using the transgenic SLI 3 and SLI 5 parasites.

2.- The choice of Pf3D7_1423700 as a control for identifying Pf-DDX5 interacting proteins in quantitative MS is questionable, since both were originally identified by ChIRP-MS in complex with RUF6 ncRNA. So even if Pf3D7_1423700 does not directly bind to RUF6 RNA, there would be the possibility that it interacts with Pf-DDX5 or shares part of the complex. Therefore, a different control

should be included to validate these results (for example GFP trap of 3D7 parasites or better parasites expressing GFP alone).

Reply: Based on two criteria, we used GFP-tagged Pf3D7_1423700 as our control. First, the tagged unknown protein did not pull down any RUF6 member. Second, no peptides for unknown function protein Pf3D7_1423700 were found in the DDX5 Pf3D7_1445900 in the Co-IP MS samples. If the 'unknown function' protein was associating with DDX5, it would have been shown in the IP MS since the samples were crosslinked to preserve the interactions with associating proteins.

3.- PfDDX5 is mostly cytoplasmic rather than nuclear in rings and also schizonts according to Fig. 3B. Please comment.

Reply: Most other eukaryotic organisms have cytoplasmic and nuclear DDX5. Wang et al. 2009 found that DDX5 (human p68) is a nucleocytoplasm shuttling protein. Interestingly, Pf-DDX5 has predicted NES (nuclear exporting signal) sequences similar to human DDX5.

4.- Table S12 should represent GO analysis of the 26 genes shared between DDX5 MS and RUF6 ChIRP listed in table S10. However, there are several genes popping up in the GO list which are not present in Table S10 (e.g. PF3D7_0320900, PF3D7_0629200).

Reply: We thank the reviewer for pointing out our error. We have corrected Table EV12.

5.- The knock-sideways data are not entirely convincing. What is the phenotype of the parasites 5 days post induction? Is the "fitness cost" due to a shift in developmental cycle or to less invasion? How does the parental parasite line grow after 5 days rapamycin treatment? The phenotype needs to be established and controlled in more detail. What is the explanation for reduction in total PfDDX5 level in Fig 5B? Please also show the cytoplasmic fraction.

Reply:

We first would like to address the point related to the cytoplasmic fraction. We have included this in the western blot below:

Figure 1: Western blot displaying Pf-DDX5 protein removal from the nucleus and cytoplasm after addition of rapamycin for two clones. Parasites were harvested at 12hpi and rapamycin was added for a total of 60 hours. Histone H3 was used as a nuclear extract control and aldolase was used as a cytoplasmic control.

To address the point related to the phenotype after 5 days of +Rapamycin treatment, we did not follow the growth of the parasites past 5 days. We did not feel it was necessary for the current work. We observed that on day 5 of the growth curve (see figure 5C), both the control and + Rapamycin were at the same stage in development (observed from giemsa staining). The +Rapamycin had a slightly lower parasitemia, possibly due to less invasion, rather than delayed growth. Additionally, Pf-DDX5 is predicted to be essential (see reference 1012-1014).

Finally, to address the point about the total fraction western blot (see figure 5B), we cannot say for sure why there is a slight difference in the amount between the – and + Rapamycin groups. We can only speculate that there might be some slight degradation as a result of the Rapamycin-induced knock sideways system.

6.- Rapamycin treatment of Pf-DDX5 for 60 hours resulted in transcriptional changes in 869 genes (app. 17% of the genome), and GO analysis would be consistent with a shift in the developmental age of the parasites (e.g. upregulation of schizont specific processes such as invasion, downregulation of ring specific processes such as translation and RNA processing). This may also affect the relative abundance of var gene transcripts. What are the transcriptional changes in the parental line after 60 hours rapamycin treatment?

Reply: In our additional supplementary Figure 3 (see our comment to reviewer 1 point 2) we show that the parasites harvested for RNA-seq were both within our synchronization time for ring stage for the -/+ Rapamycin groups.

We believe that other transcriptional changes (shown in EV14 and EV15) are the result of a lack of Pf-DDX5 (and not a difference in developmental stages) since this RNA helicase most likely has a variety of functions related to gene regulation, in addition to interacting with RUF6 ncRNA.

7.- Suppl. Figure S2: the few rings shown in a single field of view of the slides may not be representative, so a more detailed breakdown of stage composition would be critical since the transcriptional changes point to upregulation of invasion related genes which would be consistent with more residual schizonts remaining in the RAPA treated culture due to a gross growth delay. This would affect the relative level of var gene transcripts too. Ideally, var transcription should be monitored over several consecutive time points in the life cycle.

Reply: We agree with this reviewer to investigate synchronization of the Rap- and Rap+ since this is important for making the statement that var gene transcription is reduced in Rap+ (DDX5 downregulated parasites). See also our comment to reviewer 1 point 2.

The same concentration of Rapamycin was used as in Sinha et al. 2021 where they show that even after 4 days Rapamycin alone does not affect parasite growth in wildtype 3D7.

« To determine if PfYTH.2 knock-sideways leads to a functional knockdown, we compared growth rates with or without rapamycin between the PfYTH.2-sandwich and PfYTH.2-sandwich^{ML} strains. The PfYTH.2-sandwich and PfYTH.2-sandwich^{ML} strains grew at similar rates in the absence of rapamycin, and addition of rapamycin did not affect growth rates of PfYTH.2-sandwich parasites, indicating that these concentrations of rapamycin do not restrict parasite growth (Fig. 3B). »

Sinha, A., Baumgarten, S., Distiller, A., McHugh, E., Chen, P., Singh, M., Bryant, J. M., Liang, J., Cecere, G., Dedon, P. C., Preiser, P. R., Ralph, S. A., & Scherf, A. (2021). Functional Characterization of the m⁶A-Dependent Translational Modulator PfYTH.2 in the Human Malaria Parasite. *mBio*, 12(2), e00661-21. <https://doi.org/10.1128/mBio.00661-21>

8.- Is there a difference in PfEMP1 expression after Pf-DDX5 KS? This could for example be addressed by western blot with anti-ATS antibodies.

Reply: Changes to protein level might be difficult to demonstrate by Western blot since the observed decrease in var gene transcription was not very big and may lead to unconvincing quantification results. We think adding Western blot data would not add significant data that would strengthen the main message of this manuscript.

9.-The active var gene was the only deregulated var gene. What about other VSAs? According to Barcon-Simmons et al, other heterochromatic families such as PfMC2TM were also deregulated by ruf6 KD. Do you observe this as well with PfDDX5 KS?

Reply: The timepoint 12hpi was chosen to study *var* gene transcription. The other VSA peak during Trophozoite stages. No rif, stevor, nor mc-2TM genes were significantly affected in the +Rap group at the chosen time point.

We agree with the reviewer and would be interested in determining if other VSAs would be affected. More time points along the asexual blood stage cycle are necessary to explore the potential role of DDX5 for the transcription of other VSA gene families. However, it is beyond the scope of this current work since we focus specifically on the singular expression of *var* genes.

Minor comments:

10.- Introduction page 3 paragraph 1: full stop missing after citation.

Reply: Done (see line 124).

11.- Results, paragraph 1: change "...and was reproducible" to "...that was reproducible".

Reply: Done (see lines 171-172).

12.- Table S1 is missing???

Reply: we excuse for having missed to add this table to the manuscript. This has been corrected in the revised manuscript (see line 1115).

13.- Please specify which selection criteria apply to which candidate in Fig. 2D.

Reply: The table below shows which criteria applies to each candidate:

Criteria to select candidates: 1) have predicted RNA-binding potential, 2) have a function in gene activation, and/or 3) are conserved specifically in the var gene-containing Laverania species but not in malaria species that do not encode var genes (most malaria species).

Proteins	MW (kDa)	Description	Criteria met
PF3D7_0318200	278,7	DNA-directed RNA polymerase II subunit RPB1	1, 2
PF3D7_0605100	85,6	RNA-binding protein	1, 2
PF3D7_1423700	183,6	conserved Plasmodium protein, unknown function	3
PF3D7_1445900	60,0	ATP-dependent RNA helicase DDX5	1, 2
PF3D7_1023900	381,3	chromodomain-helicase-DNA-binding protein 1	2
PF3D7_1104200	167,4	chromatin remodeling protein	2
PF3D7_1002400	30,1	transformer-2 protein	1, 2
PF3D7_0819600	32,2	conserved Plasmodium protein, unknown function	3
PF3D7_0923900	23,0	RNA-binding protein	1
PF3D7_1007700	182,7	AP2 domain transcription factor AP2-I	2
PF3D7_1107300	381,9	polyadenylate-binding protein-interacting protein 1	1, 2
PF3D7_1110200	152,2	pre-mRNA-processing factor 6	2
PF3D7_1330800	68,0	RNA-binding protein	1
PF3D7_1455300	69,5	conserved Plasmodium protein, unknown function	2
PF3D7_0919000	31,8	nucleosome assembly protein	2

14.- Please provide data on validation of the transgenic parasite lines generated in this study, for example the PCR screen mentioned in the M&M section showing correct integration and lack of wt locus.

Reply: We show the requested set of data for this reviewer below and have included the primers in Table S1 (see line 1115):

SLI 3, 5 transfections PCR check

Lanes:

- 1: transfections SLI 3 WT locus check
- 2: transfections SLI 5 WT locus check
- 3: transfections SLI 3 HB F 3 SLI check R
- 4: transfections SLI 5 HB F 5 SLI check R
- 5: SLI 3 maxi prep HB F 3 SLI check R
- 6: SLI 5 maxi prep HB F 5 SLI check R
- 7: no primers gDNA control
- 8: ladder

*SLI check R includes GFP

Integrated:
 SLI 3: correct size = 1,332bp
 SLI 5: correct size = 1,606bp

SLI 5 + BSD mislocalizer transfections PCR check

Lanes:

- 1: Ladder
- 2: Clone B12
- 3: Clone A5
- 4: Clone B4
- 5: Clone A9
- 6: Negative control DSM1 plasmid
- 7: Positive control BSD maxi prep mislocalizer plasmid

Episome pLyn-FRB-mCherry BSD:
 correct size = 315bp

15.- Materials and Methods: EMSA: in vitro transcription on 0.2 g of the PCR product is probably incorrect. Please provide sequences for ruf6 probes.

Reply: We thank the reviewer for pointing out this typing mistake. The correct number is 0.2 ug of the PCR product. This has been corrected (see lines 480-481).

Primers to amplify gDNA Pf3D7_1241000 are shown here and have been added to Table S1 (see line 1115).

F: 5'-AAG-CTG-CCT-CAG-TAG-CCC-3'
R: 5'-TTG-CGC-CAC-CCC-CCT-C-3'

Pf3D7_1241000 probes:

Sense:

Biotin:AAGCUGCCUCAGUAGCCCAAUCGUUAGGUAUGUUGCCUUUCCUUGUGAGAACGUUGGUUCGACUCCGCUUG
CCGACAAUUUCAUAGGAAAAGUUGCAGAU CGAGCUGUUGGGUUACCCGGAGGGGGGCGGCGCAA

Antisense:

Biotin:UUGC GCCGCCCCCUCCGGGUAACCCAACAGCUCGAUCUGCAACUUUCCUAUGAAAUUGUCGGCAAGCGGA
GUCAACCAACGUUCACACAAGGAAAGGCAACAUACCUAACGAUUGGGCUACUGAGGCAGCUU

Sense:

AAGCUGCCUCAGUAGCCCAAUCGUUAGGUAUGUUGCCUUUCCUUGUGAGAACGUUGGUUCGACUCCGCUUGCCGA
CAAUUUCAUAGGAAAAGUUGCAGAU CGAGCUGUUGGGUUACCCGGAGGGGGGCGGCGCAA:Biotin

Antisense:

UUGC GCCGCCCCCUCCGGGUAACCCAACAGCUCGAUCUGCAACUUUCCUAUGAAAUUGUCGGCAAGCGGAGUCA
ACCAACGUUCACACAAGGAAAGGCAACAUACCUAACGAUUGGGCUACUGAGGCAGCUU:Biotin

16.- Materials and Methods: RNA IP: please specify "dilution buffer", please specify what "beads" were used.

Reply: We add the requested details to the Method section (see lines 651-654):

Dilution buffer: 10mM TRIS pH 7.5, 150mM NaCl, 0.5mM EDTA

Store at 4°C and add protease inhibitor cocktail and RNasin prior to use.

Chromotek gtma-10 GFP-Trap® Magnetic Agarose beads

17.- Materials and Methods: RNAseq and differential gene expression analysis: how were the parasites synchronized to 0 hpi? And were the 60h rapa treated parasites indeed sorbitol synchronized to 12 hpi +/- 3h again before the saponin lysis, or does this refer to synchronization at the start of rapamycin treatment?

Reply: The requested experimental details have been added to the Methods section (see lines 718-722).

18.- Materials and Methods: Co-IP-MS of GFP tagged proteins is referred to as "quantitative MS" in the text, but there seems no labelling involved. Please provide some information on normalization and data analysis, also see comments before regarding controls for this experiment.

Reply: We used label free quantitative proteomics (LC-MS/MS), as done for the ChIRP-MS (see lines 524-544 and 701-705).

October 11, 2022

RE: Life Science Alliance Manuscript #LSA-2022-01577-TR

Prof. Artur Scherf
Institut Pasteur
25, rue du Dr. Roux
Paris 75724
France

Dear Dr. Scherf,

Thank you for submitting your revised manuscript entitled "Discovery of RUF6 ncRNA interacting proteins involved in *P. falciparum* immune evasion". We would be happy to publish your paper in Life Science Alliance pending final revisions necessary to meet our formatting guidelines.

- please address Reviewer 1's remaining comments
- please upload both your main figures and your supplementary figures as single files
- please add the Twitter handle of your host institute/organization as well as your own or/and one of the authors in our system
- please add an abstract, alternate abstract/summary blurb, keywords and a category to our system
- please use the [10 author names, et al.] format in your references (i.e. limit the author names to the first 10)
- please upload your tables as editable doc or excel files or include them in the doc file of the manuscript file
- please add a callout for Figure S1 to your main manuscript text
- please provide accession info for the MS data uploaded to PRIDE
- BioProject accession number PRJNA875234 does not bring up an entry for the RNA-seq data
- please adjust your EV tables to be more Supplementary tables, and adjust the callouts in the main manuscript text

Figure Check:

- please add scale bars to Figure S2 A and B

A. FINAL FILES:

B. MANUSCRIPT ORGANIZATION AND FORMATTING:

Sincerely,

Reviewer #1 (Comments to the Authors (Required)):

Overall the authors have correctly addressed my concerns and improved the manuscript. I recommend it for publication.

The authors have correctly addressed the developmental stage question in Fig S3. The Pearson r values are low, indicating that the two populations (Rap- & Rap+) are not tightly synchronised. However what matters here is that the two populations are at the same developmental stage, and this appears to be the case.

The Discussion focuses mainly on var gene transcription regulation. However, as pointed out by Reviewer 2, a surprisingly high number of genes (869) are differentially regulated after Rapamycin treatment. Also, according to Gazanion 2020 (Plos Gen), many more G-quadruplex are found outside of var gene regions. Do the authors think that DDX5 could regulate other non-var genes through unwinding G-quadruplex?

Typo
Line 123. Full stop missing

Reviewer #2 (Comments to the Authors (Required)):

The authors have made some efforts to address the reviewer's comments. However, some key points remain only partially addressed and still leave considerable doubts about the function of DDX5 in var gene regulation.

The original RNAseq data are not accessible to this reviewer under the provided BioProject accession number. Therefore, the original data could not be assessed.

I'll reiterate some of the points made before:

What RUF6 and var genes are dominantly transcribed in the two SLI lines, particularly Pf3D7_1423700?

The use of GFP-tagged Pf3D7_1423700 as a reference in CoIPs is in the eyes of this reviewer not valid. Firstly, the lack of pull down of RUF6 RNA in RIP assays could also be due to the changed experimental conditions in this type of assay. Further, it will certainly skew the results that Pf3D7_1423700 has interaction partners itself. Consequently, the peptides identified as interacting with DDX5 are just relatively enriched to the ones interacting with Pf3D7_1423700. Actually, I'm surprised that the authors state that DDX5 was not pulled down with Pf3D7_1423700, because there are ratios for the reciprocal IPs reported in Table EV7 which would imply that it has been detected in both datasets:

The authors should really provide an unbiased GFP control, or alternatively remove this data set from the manuscript. What is the rationale for not including an unbiased control?

Line 309 – what is the evidence for rapamycin not affecting parasites after 60 h on the transcriptional level?

As pointed out previously, the data indicate a shift in the transcriptional age of the parasites, particularly as many invasion genes are upregulated after KS. Importantly, the ring stage specific gene *sbp1* (PF3D7_0501300) is downregulated to a similar level as the expressed var gene, further supporting that the observed differences are related to a shift in synchronicity or parasite age. To clearly demonstrate that the changes in var gene expression after DDX5 KD are not just the result of global changes in the transcriptional pattern of the parasites but are indeed linked to DDX5/ruf6 interactions, the authors should provide further data such as at least qPCR data demonstrating that the dominant transcript remains affected at several time points after KD. The deregulation of other VSAs may also be addressed this way.

The correlations of the RNAseq data with the microarray time course experiment (Bozdech et al) in Suppl Figure 3 appear to be relatively low (scale max $r=0.2$). There are now several high resolution RNAseq data available on PlasmoDB and in the literature to which the data should be compared (e.g. Kucharski et al, 2020 PMID: 33036628).

Please include the selection criteria in Figure 2D for clarity - they will not only be relevant for the reviewers.

Reviewer #1 (Comments to the Authors (Required)):

Overall the authors have correctly addressed my concerns and improved the manuscript. I recommend it for publication.

The authors have correctly addressed the developmental stage question in Fig S3. The Pearson r values are low, indicating that the two populations (Rap- & Rap+) are not tightly synchronised. However what matters here is that the two populations are at the same developmental stage, and this appears to be the case.

The Discussion focuses mainly on *var* gene transcription regulation. However, as pointed out by Reviewer 2, a surprisingly high number of genes (869) are differentially regulated after Rapamycin treatment. Also, according to Gazanion 2020 (Plos Gen), many more G-quadruplex are found outside of *var* gene regions. Do the authors think that DDX5 could regulate other non-*var* genes through unwinding G-quadruplex?

We thank the reviewer for bringing up this point. Based on our RNA-seq study when Pf-DDX5 was decreased in the nucleus, we assume that Pf-DDX5 could be involved in resolving G-quadruplexes in other genes of the parasite. This could explain the observed decrease in transcription of certain genes. Validating this point would need more in depth experiments.

Typo

Line 123. Full stop missing

This has been corrected.

Reviewer #2

The authors have made some efforts to address the reviewer's comments. However, some key points remain only partially addressed and still leave considerable doubts about the function of DDX5 in *var* gene regulation.

- The original RNAseq data are not accessible to this reviewer under the provided BioProject accession number. Therefore, the original data could not be assessed.

This has been corrected. The data are now available. The Mass Spec data is also available: Submission details:

Project Name: Discovery of RUF6 ncRNA interacting proteins involved in *P. falciparum* immune evasion

Project accession: PXD036801

Project DOI: Not applicable

Reviewer account details:

Username: reviewer_pxd036801@ebi.ac.uk

Password: nUBID6Yr

I'll reiterate some of the points made before:

- What RUF6 and *var* genes are dominantly transcribed in the two SLI lines, particularly Pf3D7_1423700?

These parasite lines were not cloned therefore there is no dominantly transcribed *var* gene nor RUF6 (both are clonally variant gene families). Additionally, as stated previously, the q-PCR primers for RUF6 will amplify all members equally due to the highly conserved sequence between RUF6 members. For this reason, we will detect any RUF6 family member in our RIP experiment without introducing a bias. For this reason, we think that the question of the reviewer is not relevant for the work presented in this manuscript.

- The use of GFP-tagged Pf3D7_1423700 as a reference in CoIPs is in the eyes of this reviewer not valid. Firstly, the lack of pull down of RUF6 RNA in RIP assays could also be due to the changed experimental conditions in this type of assay. Further, it will certainly skew the results that Pf3D7_1423700 has interaction partners itself. Consequently, the peptides identified as interacting with DDX5 are just relatively enriched to the ones interacting with Pf3D7_1423700. Actually, I'm surprised that the authors state that DDX5 was not pulled down with Pf3D7_1423700, because there are ratios for the reciprocal IPs reported in Table EV7 which would imply that it has been detected in both datasets:

Accession No	SLI5 Pf3D7/SLI3 Pf3D7 - Control						Total peptides in set	MW (kDa)	Description
	Ratio	Log2(Ratio)	Adj. p-value	CV %	Dist. pept. used	Pept. used			
Pf3D7_1423	0,33303049	-1,5862738	1,12757E-14	8,12591658	7	52	59	183,6	conserved Plasmodium protein, unknown function
Pf3D7_1445	4,87950791	2,28673566	3,91552E-28	4,7024762	9	71	76	60,0	ATP-dependent RNA helicase DDX5, putative

The authors should really provide an unbiased GFP control, or alternatively remove this data set from the manuscript. What is the rationale for **not** including an unbiased control?

We believe that the reviewer does not fully understand how to interpret the quantifications from MS. A negative $\log_2(\text{ratio})$ in the table above means that the control protein (Pf3D7_1423700) was ONLY significantly enriched in the control group, NOT in the DDX5 group. The same goes for DDX5, DDX5 was ONLY significantly enriched in the DDX5 group, NOT the control group. The MS quantifications clearly show that the two proteins (Pf3D7_1423700 and DDX5 Pf3D7_1445900) are not pulled together in either of the IPs.

Additionally, we provide IFA showing that Pf3D7_1423700 is trafficked out of the parasite most likely the parasitophorous vacuole with no detectable nuclear location. Therefore, our experimental evidence fully supports the use of GFP-tagged Pf3D7_1423700 as a reference.

Pf3D7_1423700

Fig. 1. Localisation of the GFP-tagged Pf3D7_1423700 protein in the infected erythrocyte. Dapi (blue) shows the pathogen nucleus. The tagged protein is shown in green.

- Line 309 – what is the evidence for rapamycin not affecting parasites after 60 h **on the transcriptional level?**

The knock-sideways system is widely used in *Plasmodium falciparum* where rapamycin is used solely to induce the dimerization of FKBP and FRB to mislocalize a protein of interest away from its site of action. No transcriptional changes have been reported in *P. falciparum* asexual blood stages in the literature for extended use of rapamycin concentrations used in our experiments.

- As pointed out previously, the data indicate a shift in the transcriptional age of the parasites, particularly as many invasion genes are upregulated after KS. Importantly, the ring stage specific gene *sbp1* (PF3D7_0501300) is downregulated to a similar level as the expressed var gene, further supporting that the observed differences are related to a shift in synchronicity or parasite age. To clearly demonstrate that the changes in var gene expression after DDX5 KD are not just the result of global changes in the transcriptional pattern of the parasites but are indeed linked to DDX5/ruf6 interactions, the authors should provide further data such as at least qPCR data demonstrating that the dominant transcript remains affected at several time points after KD. The deregulation of other VSAs may also be addressed this way.
- The correlations of the RNAseq data with the microarray time course experiment (Bozdech et al) in Suppl Figure 3 appear to be relatively low (scale max $r=0.2$). There are now several high resolution RNAseq data available on PlasmoDB and in the literature to which the data should be compared (e.g. Kucharski et al, 2020 PMID: 33036628).

We have shown now with 4 different datasets of published reference RNA-seq transcription studies (including the proposed one by this reviewer) that there is no difference in cell cycle

progression with our treatment group. Based on these data, we do not believe that qPCR is necessary to back up RNA-sequencing data. We have now included the following 3 additional correlations. All of which show that there is no difference in cell cycle progression between the control and treatment group.

Kucharski M, Tripathi J, Nayak S, Zhu L, Wirjanata G, van der Pluijm RW, Dhorda M, Dondorp A, Bozdech Z. A comprehensive RNA handling and transcriptomics guide for high-throughput processing of Plasmodium blood-stage samples. *Malar J.* 2020 Oct 9;19(1):363.

Subudhi AK, O'Donnell AJ, Ramaprasad A, Abkallo HM, Kaushik A, Ansari HR, Abdel-Haleem AM, Ben Rached F, Kaneko O, Culleton R, Reece SE, Pain A. Malaria parasites regulate intra-erythrocytic development duration via serpentine receptor 10 to coordinate with host rhythms. *Nat Commun.* 2020 Jun 2;11(1):2763.

Painter, H.J., Chung, N.C., Sebastian, A. *et al.* Genome-wide real-time in vivo transcriptional dynamics during *Plasmodium falciparum* blood-stage development. *Nat Commun* 9, 2656 (2018).

Additionally, we looked at ring stage genes used to estimate parasite age:
 Ciuffreda, L., Zoiku, F. K., Quashie, N. B., & Ranford-Cartwright, L. C. (2020). Estimation of parasite age and synchrony status in *Plasmodium falciparum* infections. *Scientific reports*, 10(1), 10925.

We found no significant difference between the control and treatment group.

- Please include the selection criteria in Figure 2D for clarity - they will not only be relevant for the reviewers.

We have added this information to the **supplementary data, Table S7**.

October 26, 2022

RE: Life Science Alliance Manuscript #LSA-2022-01577-TRR

Prof. Artur Scherf
Institut Pasteur
25, rue du Dr. Roux
Paris 75724
France

Dear Dr. Scherf,

Thank you for submitting your Research Article entitled "Discovery of RUF6 ncRNA interacting proteins involved in *P. falciparum* immune evasion". It is a pleasure to let you know that your manuscript is now accepted for publication in Life Science Alliance. Congratulations on this interesting work.

DISTRIBUTION OF MATERIALS:

Again, congratulations on a very nice paper. I hope you found the review process to be constructive and are pleased with how the manuscript was handled editorially. We look forward to future exciting submissions from your lab.

Sincerely,
